# Unveiling Causal Reasoning in Large Language Models: Reality or Mirage?

**Haoang Chi**[1,2]* , **He Li**[2]* , **Wenjing Yang**[2]† , **Feng Liu**[3] , **Long Lan**[2] , **Xiaoguang Ren**[1] ,
**Tongliang Liu**[4] , **Bo Han**[5]

[1] Intelligent Game and Decision Lab,  [2] National University of Defense Technology,
[3] University of Melbourne,  [4] University of Sydney,  [5] Hong Kong Baptist University
{haoangchi618,fengliu.ml}@gmail.com, rxg_nudt@126.com
{lihe_117,wenjing.yang, long.lan}@nudt.edu.cn,
tongliang.liu@sydney.edu.au, bhanml@comp.hkbu.edu.hk

## Abstract

Causal reasoning capability is critical in advancing large language models (LLMs) toward strong artificial intelligence. While versatile LLMs appear to have demonstrated capabilities in understanding contextual causality and providing responses that obey the laws of causality, it remains unclear whether they perform genuine causal reasoning akin to humans. However, current evidence indicates the contrary. Specifically, LLMs are only capable of performing shallow (*level*-1) causal reasoning, primarily attributed to the causal knowledge embedded in their parameters, but they lack the capacity for genuine human-like (*level*-2) causal reasoning. To support this hypothesis, methodologically, we delve into the autoregression mechanism of transformer-based LLMs, revealing that it is not inherently causal. Empirically, we introduce a new causal Q&A benchmark called CausalProbe-2024, whose corpora are fresh and nearly unseen for the studied LLMs. The LLMs exhibit a significant performance drop on CausalProbe-2024 compared to earlier benchmarks, indicating the fact that they primarily engage in *level*-1 causal reasoning. To bridge the gap towards *level*-2 causal reasoning, we draw inspiration from the fact that human reasoning is usually facilitated by general knowledge and intended goals. We propose $G^2$-Reasoner, a method that incorporates general knowledge and goal-oriented prompts into LLMs' causal reasoning processes. Experiments demonstrate that $G^2$-Reasoner significantly enhances LLMs' causal reasoning capability, particularly in fresh and counterfactual contexts. This work sheds light on a new path for LLMs to advance towards genuine causal reasoning, going beyond *level*-1 and making strides towards *level*-2.

## 1   Introduction

The emergent of large language models (LLMs), such as GPT 4 [45], Gemini 1.5 [16], and Claude 3 [2], have significantly changed the paradigm of how people work and do research in recent years, demonstrating competitive abilities (LLMs) in instruction following [1, 7, 45, 60], in-context learning [8, 12, 67], reasoning [63, 70], coding [10, 18, 51] and etc. LLMs have demonstrated remarkable abilities in processing and generating human-like text, leading to a belief that they may possess the intelligence akin to human cognition. Reasoning is an essential component of human intelligence and a prominent characteristic that distinguishes humans from other species [47]. Many recent works

---

*Equal contribution
†Corresponding author

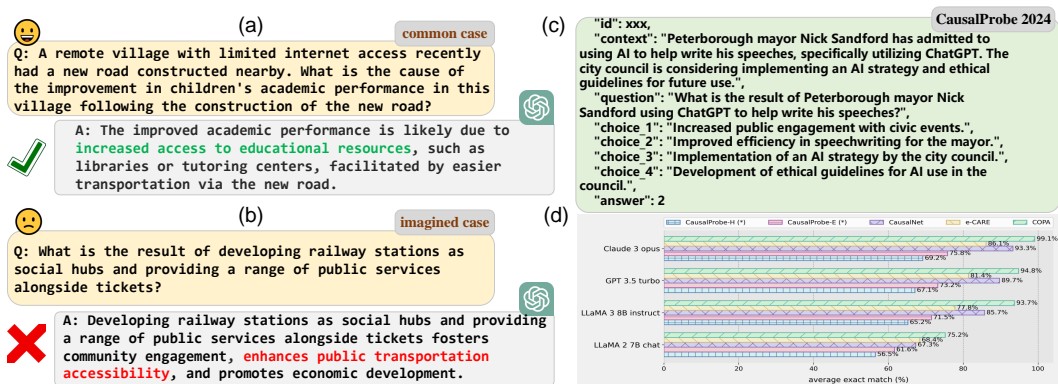

Figure 1: The motivation of this work. (a) LLMs work well on common causal reasoning tasks, whose topics usually are widely discussed. (b) LLMs struggle to tackle rare tasks, whose corpora are possibly brand new for them. (c) Here is an example of the CausalProbe 2024 benchmark that is introduced to examine the true level of causal reasoning in LLMs, including an easy one-choice version (CausalProbe-E), a hard one-choice version (CausalProbe-H), and an uncertain multiple-choice version (CausalProbe-M). They have analogous formats but different construction strategies. (d) Compared to previous causal Q&A benchmarks, the studied LLMs exhibit a significant performance drop on CausalProbe 2024. (*) represents our benchmarks.

have dived into evaluating and improving LLMs' general reasoning capabilities, such as logical reasoning [11, 20] and mathematical reasoning [22, 57].

In this work, we focus on causal reasoning, an advanced reasoning form [31]. In the context of LLMs, causal reasoning is to discern the cause-and-effect relationships that govern the physical world from a text [43, 54]. Current LLMs appear to have demonstrated some degree of causal reasoning capabilities [73]. Sometimes, when asked about the cause or effect of a given text, LLMs can accurately provide responses satisfying the laws of causality that originate from the physical world. For example, in Figure 1(a), the LLM can work out reasonable causes (i.e., "increased access to educational resources") for academic performance improvement. Faced with such encouraging performances, we have to pose a question:

> *Does this reflect LLMs' genuine causal reasoning capability or only a "mirage"?*

The answer leans more towards the latter. We find that LLMs are adept at qualitatively solving causal reasoning tasks related to common knowledge (e.g., Figure 1 (a)), but they struggle to tackle more advanced task types (e.g., Figure 1 (b)), such as discovering new causal knowledge and estimating specific causal quantities. For example, in Figure 1(b), when asked about the effect of an imagined, unusual case ("developing railway stations as social hubs..."), the LLM's answer ("enhance public transportation accessibility") is clearly irrelevant with such an action. Recent works [64, 72, 73] also came to a similar conclusion. Given the performance differences on tasks of varying difficulty, we propose a hypothesis: the apparent (*level*-1) causal reasoning capabilities of LLMs can be primarily attributed to associated knowledge from their training corpora, rather than engaging in genuine, human-like (*level*-2) reasoning. We will justify this hypothesis from two aspects.

From a methodological perspective, the widely-used transformer-based LLMs essentially perform next-token prediction [35, 61], which is realized in an autoregressive manner. Autoregressive model [71] originates from time-series analysis with an important assumption: the current value is determined by past values, not related to future values. For texts, however, the fact that the current token depends on the past tokens does not necessarily mean that there is a causal relationship between them. In addition, philosopher David Hume raised a viewpoint that sequential causality is not equivalent to logical causality [21]. Therefore, this mechanism makes LLMs good at reusing the causal knowledge in their training corpora but often makes them struggle to comprehend and generate texts that capture genuine causal knowledge. We also use structural causal models (SCMs) to formally account for this intuitive conclusion (see Section 4.1).

From an empirical perspective, to validate the hypothesis that LLMs only possess *level*-1 causal reasoning capabilities, we introduce a new causal question & answer (Q&A) benchmark (Figure

1(c)), named CausalProbe 2024, whose corpora was made public later than the release of the studied LLMs.[3] Given that the training data is rarely disclosed, we can, at a minimum, guarantee that the corpora of CausalProbe 2024 is not included verbatim in the training data of the studied LLMs. Compared to earlier causal Q&A benchmarks, such as COPA [48], e-CARE [14] and CausalNet [4], all the studied LLMs (e.g., LLaMA 2 7B, LLaMA 3 8B, GPT 3.5 turbo, Claude 3 Opus) exhibit a significant performance drop on CausalProbe 2024 (Figure 1(d)). As the earlier benchmarks are potentially part of the training data, these longitudinal comparisons largely support our hypothesis.

Existing studies about LLMs' causal reasoning mainly focused on only assessments [15, 26, 49] and designing prompt-based approaches [4, 25]. These studies have taken a significant step forward in advancing LLMs' causal reasoning. However, they overlooked two basic principles of human reasoning: general knowledge and intention. For example, when we reason about a mathematical problem, we always take the basic axioms as a reference, with the ultimate goal as guidance. Inspired by this fact, we propose a causal reasoning framework for LLMs called $G^2$-Reasoner, which incorporates general knowledge and goal-oriented prompts during reasoning. Specifically, we use the retrieval-augmented generation (RAG) to incorporate external knowledge bases. Then we stimulate the LLMs to consistently discern correct causal relationships in contexts to reach the final responses. Experiments show that $G^2$-Reasoner significantly enhances LLMs' causal reasoning capabilities towards *level*-2, especially on fresh even fictitious tasks (e.g., CausalProbe 2024), which is consistent for both open-source and closed-source LLMs.

## 2 Related Work

In this section, we review the related works, including LLMs' reasoning, LLMs' causal reasoning, and LLMs' causal reasoning benchmarks. In Appendix D, we discuss the related works in detail.

### 2.1 Reasoning in Large Language Models

Reasoning ability is crucial to LLMs' performance on tasks such as theorem proving, problem-solving, and robotics [41, 58]. [58] propose a comprehensive review of the foundation models' reasoning, they discuss reasoning tasks including commonsense reasoning, mathematical reasoning, logical reasoning, and causal reasoning. Commonsense reasoning refers to the reasoning process that utilizes commonsense and daily life experiences [9]. Several commonsense question-answering datasets have been proposed to test LLMs' commonsense reasoning ability [19, 29, 74, 75]. Causal reasoning is the process of identifying and understanding the cause-and-effect relationships between variables or events, which involves identifying potential causes and effects within a system or context [58]. One distinctive feature of causal reasoning is that it involves counterfactual reasoning, which means reasoning within a hypothetical scenario. Our work focuses on the LLMs' causal reasoning, particularly counterfactual reasoning.

### 2.2 Causal Reasoning in Large Language Models

Causal Reasoning tasks include causal discovery, cause attribution, and causal effect estimation [30, 58, 65]. Causal discovery aims to recover the latent causal structure of variables. Cause attribution refers to uncovering potential causes behind a process, while causal effect estimation aims to investigate the effect of cause variables [26, 76]. While LLMs already show the ability to uncover causal relationships from context, their abilities have some limitations [37, 4, 39, 77]. Counterfactual reasoning is an essential task of causal reasoning. The difference between counterfactual reasoning and other causal tasks is counterfactual reasoning involves reasoning in hypothetical scenarios [28, 58]. Several studies conclude that LLMs have limitations when encountering hypothetical scenarios in counterfactual reasoning [33, 66].

---

[3]To double ensure the freshness of the corpora in CausalProbe 2024, we use a membership inference attack approach (i.e., determining whether an arbitrary sample is part of a given LLM's training data), Min-$K$% Prob [53], to evaluate it and earlier benchmarks. The results are shown in Section 6.1.

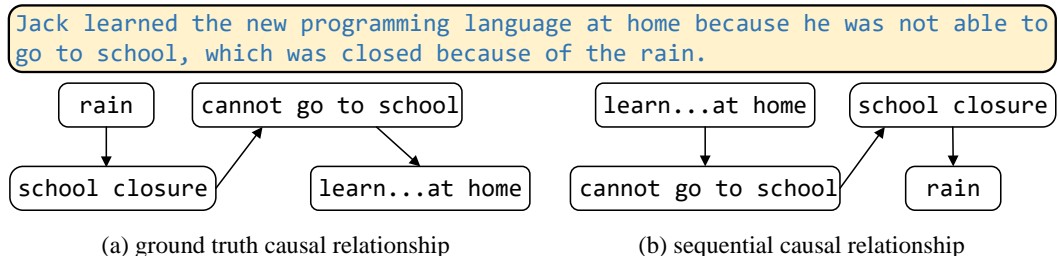

(a) ground truth causal relationship            (b) sequential causal relationship

Figure 2: An diagram of illustrating how autoregression fails to capture the correct causal knowledge.

## 2.3 Causal Reasoning Benchmarks in Large Language Models

There have been extensive studies on causal reasoning benchmarks. Existing causal reasoning benchmarks are mainly causal question-answering datasets. [6] employs language rules to extract causal questions from ten large question-answering datasets to form the CausalQA. CRAB [50] is a dataset that aims to assess LLMs' abilities to understand causal relationships among real-world events. FCR [69] is a human-labeled dataset that includes 24K question-answering pairs. Cladder [57] is a dataset that involves symbolic questions and corresponding ground truth answers, [57] employs causal graphs and structural causal models to generate the dataset. CausalProbe 2024 is different from the above benchmarks, as its contents are based on the latest and authoritative information, which is unlikely to be encompassed by the pre-training corpora of LLMs.

## 3 Problem Formalization

In this section, we introduce and clarify the necessary definitions used in this work. First, we provide a formal definition for causal reasoning in the context of LLMs. Then, we introduce two levels of causal reasoning capability to reveal the limitations of LLMs in this aspect. Last, we use causal language to depict the causal reasoning of LLMs.

In this work, the scope of causal reasoning of LLMs we study is reasoning about causal knowledge in textual form, distinguish from the numerical form in statistical causal inference [23].

**Definition 1** (causal reasoning in LLMs). *In the context of large language models, the causal reasoning consists of two aspects:*

- *comprehend the given contexts and discern the causal relationship within them;*
- *responds to the causality-related queries, obeying the contexts and objective laws of causality.*

To reach the major conclusions of this work, we first categorize LLMs' causal reasoning capability into two levels, motivated by the results of cognition science [56] and causality science [17].

**Definition 2** (*level*-1 causal reasoning). *Level-1 causal reasoning involves retrieving causal knowledge embedded in model parameters and contextual information. This form of reasoning is typically fast and well-suited for handling simple cause-and-effect relationships.*

**Definition 3** (*level*-2 causal reasoning). *Level-2 causal reasoning leverages sophisticated reasoning mechanisms and internal parametric knowledge and contexts to deduce causal knowledge, including new/unseen causal knowledge. This form of reasoning is typically slow and capable of deriving new causal knowledge.*

The above two definitions are inspired by 'Thinking, Fast and Slow' [27]. *Level*-2 causal reasoning is not necessarily always better than *level*-1. Ideally, the interplay and adaptive switching and of these two levels of causal reasoning are crucial for LLMs to work both rapidly and reliably. In this work, we aim to explore the causal reasoning capabilities of current LLMs in terms of these two levels and dig into the underlying reasons.

**Remark 1.** *In this work, we only consider a simple type of causal reasoning tasks that contain a single cause-effect pair. The cases of multiple cause-effect pairs and mediators are excluded. In addition, we primarily consider qualitative causal reasoning tasks (e.g., causal discovery [4]), rather than quantitative ones (e.g., treatment effect estimation [25]). In summary, the causal reasoning tasks that we focus on can be categorized into two type: 1) what is the reason $\cdots$; 2) what is the result $\cdots$. Due to the complexity of natural language, there are many different sentence patterns to express it.*

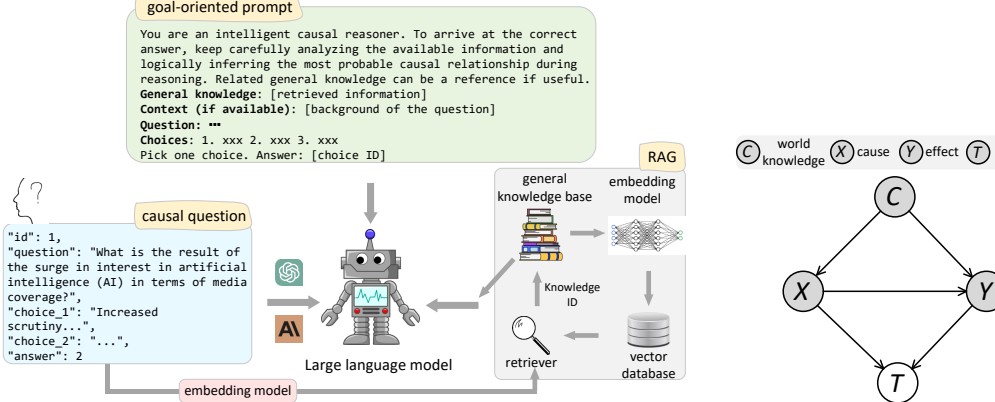

Figure 3: The diagram of G²-Reasoner. This framework consists of two modules. One module is a retrieval-augmented generation (RAG) system to retrieve general knowledge that is related to the causal question. Another is a goal-oriented prompt to steer LLMs to race toward the ultimate goal of causal reasoning.

Figure 4: The causal graph that depicts the data generation mechanism of causal reasoning in LLMs.

## 4 LLMs cannot Perform Genuine Causal Reasoning

In this section, we aim to explore the real causal reasoning capability of current LLMs, in terms of our pre-defined two capability levels: *level*-1 (Definition 2) and *level*-2 (Definition 3). We study this problem from both a methodological perspective and an empirical perspective.

### 4.1 Autoregressive LLMs are not Necessarily Causal

Although autoregressive LLMs have achieved great successes, recently, they began to be doubted by this community [34, 42]. We aim to study why the autoregressive mechanism prevents LLMs from acquiring *level*-2 causal reasoning capabilities. Essentially, the decoder-only transformer-based LLMs that are widely used today to perform next-token prediction, are trained with an autoregressive loss [35]. In statistics, autoregression model is based on a fundamental assumption: in a sequence, the current value is determined by past values, not related to future values. However, causal knowledge expressed through natural languages does not necessarily satisfy this assumption. This is because sequential causality is not equivalent to logical causality [21]. For example, due to the variability of natural language, sentence patterns may lead to false sequential causal relationships, and Figure 2 shows a toy instance. This sentence consists of four concepts, i.e., "*rain*", "*school closure*", "*cannot go to school*", and "*learn the new programming language at home*". However, we can easily find that the sequential causal relationship in this sentence is totally incorrect. Thus, autoregressive LLMs suffer from capturing logical causal knowledge in complex texts, restricting their generalization abilities on unseen tasks. Note that due to the vast training corpus of LLMs, they may have encountered different textual expressions with the same semantic meaning, thus still being able to infer correct causal relationships in some complex tasks.

Assume that $c = (c_1, c_2, \ldots, c_k)$ is a tokenized context and $(w_1, w_2, \ldots, w_t)$ are already generated tokens, LLMs can obtain a distribution of the next token $w_{t+1}$: $P(w_{t+1}|c, w_1, w_2, \ldots, w_t)$, which is usually obtained by a softmax function. Then, LLMs predict the next token from this distribution in a sampling method, such as greedy sampling and beam search. Autoregressive training makes LLMs memorize common causal knowledge expressions. Based on the above discussion, there are two major issues: 1) If the context $c$ is not sequentially causal and unfamiliar for LLMs, they tend to misunderstand the causal knowledge in $c$; 2) If $P(w_{t+1}^*|c, w_1, w_2, \ldots, w_t)$ is large[4] but the text represented by $(\ldots, w_{t-1}, w_t, w_{t+1}^*)$ is inconsistent with the laws of causality or the context $c$, LLMs tend to respond an incorrect causal reasoning result. In contrast, if the context $c$ and generated $(\ldots, w_{t-1}, w_t, w_{t+1}^*)$ express correct causal knowledge and are familiar for LLMs, LLMs will perfectly address this task. Thus, the autoregression mechanism makes LLMs' causal reasoning primarily rely on correct causal knowledge in a large number of training corpora, i.e., the *level*-1 causal reasoning. In the following, we will validate this hypothesis from an empirical perspective.

---

[4]Probably because the token sequence $(\ldots, w_{t-1}, w_t, w_{t+1}^*)$ appears frequently in the training corpora.

Table 1: The data cut-off time comparison of the studied LLMs and CausalProbe 2024 benchmark.

| LLaMA 2 7B chat | LLaMA 3 8B instruct | GPT 3.5 turbo | Claude 3 opus | CausalProbe 2024 |
|---|---|---|---|---|
| Sep 2022 | Mar 2023 | Sep 2021 | Aug 2023 | Jan 2024 |

## 4.2 Empirical Study on Causal Reasoning Capabilities of LLMs

In this section, we study the capability boundary of LLMs' causal reasoning for causal reasoning. Recall the hypothesis that LLMs are only capable of performing *level*-1 reasoning, primarily attributed to the causal knowledge embedded in their parameters during training, but struggle to perform genuine causal reasoning in complex or uncommon cases. Therefore, in order to physically avoid LLMs seeing the same corpora as the training data, we introduce a new causal question & answer (Q&A) benchmark, named CausalProbe 2024 (Figure 1(c)). The corpora used to construct CausalProbe 2024 are published later than January 1, 2024, which is later than the training data cut-off times of all the studied LLMs (Table 1). Thus, the studied LLMs should not have seen corpora that is the same or very similar to CausalProbe 2024. The construction details of CausalProbe 2024 are presented in Section 6.1. CausalProbe 2024 consists of three datasets: CausalProbe 2024 Easy (*abbr.*, CausalProbe-E), CausalProbe 2024 Hard (*abbr.*, CausalProbe-H), and CausalProbe 2024 Multiple-choice (*abbr.*, CausalProbe-M). These three datasets progressively probe whether LLMs can solve novel causal reasoning tasks, whether they can distinguish misleading false causal relationships, and whether they can reason about multiple causal relationships within a scenario.

To validate whether LLMs heavily rely on the causal knowledge embedded in their parameters, we employ three earlier causal Q&A benchmarks as a comparison, i.e., COPA [48], e-CARE [14], and CausalNet [4]. The corpora of COPA is released earlier than 2011. An example of their comparison is shown in Figure 9. The corpora of e-CARE (i.e., GenericsKB [5]) is released earlier than 2020. CausalNet is also a fresh causal Q&A benchmark and its corpora are generated by ChatGPT, serving as an intermediate comparison. The corpora of COPA and e-CARE are likely to exactly be the training data of the studied models. For CausalNet, we have reason to speculate that its corpora may be similar to the training data of ChatGPT, although a large temperature hyperparameter can bring some creativity. Their detailed introduction in presented in Appendix F. In terms of the format, CausalProbe 2024 is consistent with them, including *ID*, *question*, *choices*, and *answer*. In addition, CausalProbe 2024 additionally provides contexts as the background knowledge of questions. In Section 6.1, we discuss the reasonability of providing contexts rather than only providing questions.[5] To doubly guarantee the freshness of CausalProbe 2024, we employ a LLM's membership inference attack method, Min-$K$% Prob [53], to detect how much the corpora of a benchmark potentially comes from a LLM's training data. The results are shown in Table 3, showing that the corpora of CausalProbe 2024 are fresher for the studied LLMs than other benchmarks.

From Figure 1(d), we can observe significant performance drops on CausalProbe-E and CausalProbe-H for all four studied LLMs. Especially for CausalProbe-H, Claude 3 opus, a current SOTA LLM, only achieves the average exact match less than 70%. The popular and competitive open-source LLM, LLaMA 2 7B chat, just get it half right. As the corpora of CausalProbe 2024 comes from news, it is close to everyday life and hardly consists of professional concepts and unfamiliar words. Thus, the main cause of performance degradation is the freshness of corpora, indicating the fact that LLMs only are capable of doing *level*-1 causal reasoning, instead of genuine *level*-2 causal reasoning. The empirical result is in perfect agreement with the analysis derived from the autoregressive mechanism.

## 5 G$^2$-Reasoner: A General-Knowledge-Assisted and Goal-Driven Reasoner

Until now, we have discovered of the limitation of LLMs' causal reasoning capabilities from both methodological and empirical perspectives. To move towards *level*-2 causal reasoning, we seek inspiration from human beings. Human reasoning processes, including causal reasoning, are driven by specific reasoning tasks and supported by extensive foundational knowledge acquired throughout life, following established reasoning patterns such as deductive and inductive reasoning [44]. When

---

[5]In COPA, e-ECARE, and CausalNet, their data also contain "context" or "premise". However, they are not real background knowledge, but a part of the questions themselves. In CausalProbe 2024, the "question" is equivalent to "context" plus "question" in the other three benchmarks.

solving plane geometry proof problems, students apply three basic axioms as criteria while working toward the target proposition. Unlike humans, LLMs perform causal reasoning through next-token predictions based solely on patterns learned during training, without necessary knowledge to guide their reasoning. Drawing inspiration from causal inference, we formalize textual causal reasoning tasks[6] using a causal graph, as shown below.

**Causal model for causal reasoning in LLMs.** Based on Definition 1, we use a causal graph (Figure 4) to formally depict the textual causal reasoning task of LLMs. Based on this formalization, we can identify the key factors for LLMs to perform causal reasoning well. Given a cause semantic variable $X$ and an effect semantic variable $Y$, naturally, we have $X \rightarrow Y$. The fact that $X$ causing $Y$ is driven by the laws of the physical world or imagined/virtual worlds $C$. Then, the semantic variables $X$ and $Y$ are transformed into a variable $T$ that represents the natural language, through a mapping $h$, i.e., $h(X, Y, \epsilon) = T$, where $\epsilon$ is a random exogenous variable. In this formalization, $h$ can be viewed as a language system to transform two causal concepts into a paragraph of text. The variable $\epsilon$ represents various factors in generating readable text from causal concepts $X$ and $Y$, such as language type, context, and mode of expression (e.g., active or passive voice). While linguistic variability ($\epsilon$) enriches the diversity and flexibility of natural language, it poses challenges for LLMs' causal reasoning. Consider the causal relationship between "smoking" ($X$) and "lung cancer" ($Y$). This relationship can be expressed in various ways, such as: 1) "A history of smoking is a common risk factor for lung cancer," and 2) "Knowing that smoking greatly increases the risk of lung cancer, why take the risk?" Although these sentences convey the same underlying causal relationship, they differ significantly in their linguistic structure and expression (i.e., different $\epsilon$). The natural language expression $T$ encapsulates both the fundamental causal relationship between $X$ and $Y$, as well as the contextual nuances and linguistic style represented by $\epsilon$.

We can easily find that the causal graph (Figure 4) is exactly a causal model with a confounding variable and a conditioned collider [46]. $T$ is conditioned because the observed natural language descriptions in textual causal reasoning tasks are inherently deterministic. Following a fundamental principle of causal inference, when the collider $T$ is conditioned ($T = T_0$), it creates an association between $X$ and $Y$. Thus, natural language descriptions of causal reasoning tasks provide valuable information for determining causal relationships.[7] For cause-to-effect tasks, given $P_Y$ (the distribution of possible effects $Y$ generated by LLMs), our objective is as follows:

$$\arg \max_{Y \sim P_Y} \mathbb{P}[Y | X = X_0, T = T_0, C] \tag{1}$$

where $X_0$ is the cause described in a causal reasoning question. However, to account for the confounding variable $C$, we can apply the total probability formula, i.e.,

$$\arg \max_{Y \sim P_Y} \mathbb{E}_{C \sim P_C} \mathbb{P}[Y | X = X_0, T = T_0, C] = \arg \max_{Y \sim P_Y} \mathbb{P}[Y | X = X_0, T = T_0], \tag{2}$$

where $P_C$ is the general knowledge base. While strictly ensuring the validity of Eq. (2) would require a complete general knowledge base, which is impractical, Eq. (2) nonetheless provides an insightful approach to addressing causal reasoning problems in LLMs and is helpful in practice (Section 6.3).

**Goal-driven prompt.** As discussed in Section 4.1, the autoregressive nature of LLMs hinders their understanding of correct causal relationships. As generated sequences lengthen, LLMs tend to lose coherence and deviate from their initial targets [40]. To maintain focused generation, we design a goal-driven prompt that guides LLMs in identifying correct causal relationships during the decoding process, as shown in Figure 3.

Motivated by human reasoning mechanism and causal graph theory, we propose a novel causal reasoning framework (Figure 3), named $G^2$-Reasoner, which involves general knowledge as a basis and intended goals as a guide. Specifically, $G^2$-Reasoner leverages a small ($\sim$16 Mb) general knowledge Q&A dataset[8] as the knowledge base, enabling the model to draw upon related knowledge

---

[6]In this work, we focus on textual causal reasoning tasks, instead of the numerical ones like classical causal inference/discovery.

[7]Note that collider $T$ is useful in our problem, although common causal inference tasks treat it as a bias.

[8]General knowledge dataset: https://huggingface.co/datasets/MuskumPillerum/General-Knowledge. We only use its answers as the retrieval document, because the answers are declarative sentences that are more suitable to construct a knowledge base. In addition, the answers have contained the entity information in the questions.

Table 2: Results of the studied LLMs on four causal Q&A benchmarks. The metric is exact match (EM). "Vanilla" denotes doing inference directly. "C-E", "C-H" represnt CausalProbe-E and CausalProbe-H. The standard deviations are presented in Appendix G.

| | | LLaMA 2 | LLaMA 3 | GPT 3.5 | Claude 3 |
|---|---|---|---|---|---|
| COPA | Vanilla | 0.752 | 0.937 | 0.948 | 0.991 |
| | CoT | 0.812 | 0.944 | 0.951 | 0.991 |
| | RAG | 0.757 | 0.912 | 0.936 | 0.990 |
| | $G^2$-Reasoner | 0.813 | 0.948 | 0.953 | 0.990 |
| e-CARE | Vanilla | 0.684 | 0.778 | 0.814 | 0.861 |
| | CoT | 0.697 | 0.770 | 0.802 | 0.864 |
| | RAG | 0.687 | 0.760 | 0.809 | 0.836 |
| | $G^2$-Reasoner | 0.701 | 0.779 | 0.821 | 0.849 |
| CausalNet | Vanilla | 0.673 | 0.857 | 0.897 | 0.933 |
| | CoT | 0.666 | 0.767 | 0.874 | 0.910 |
| | RAG | 0.650 | 0.860 | 0.898 | 0.922 |
| | $G^2$-Reasoner | 0.681 | 0.855 | 0.898 | 0.929 |
| C-E (*) | Vanilla | 0.616 | 0.715 | 0.732 | 0.758 |
| | CoT | 0.636 | 0.720 | 0.737 | 0.753 |
| | RAG | 0.621 | 0.704 | 0.741 | 0.756 |
| | $G^2$-Reasoner | 0.642 | 0.718 | 0.746 | 0.758 |
| C-H (*) | Vanilla | 0.565 | 0.652 | 0.671 | 0.692 |
| | CoT | 0.573 | 0.644 | 0.667 | 0.701 |
| | RAG | 0.575 | 0.655 | 0.678 | 0.685 |
| | $G^2$-Reasoner | 0.582 | 0.658 | 0.693 | 0.696 |

Table 3: Results of training data detection using Min-$K$% Prob [53]. We conduct this evaluation on LLaMA 2 7B and LLaMA 3 8B. The metric is the average negative log-likelihood on a dataset. A smaller value indicates better freshness. "C-E", "C-H" represnt CausalProbe-E and CausalProbe-H.

| | | LLaMA 2 | LLaMA 3 |
|---|---|---|---|
| COPA | Min-10% Prob | 13.27 | 16.64 |
| | Min-20% Prob | 10.57 | 12.21 |
| | Min-30% Prob | 8.97 | 10.32 |
| e-CARE | Min-10% Prob | 13.08 | 14.48 |
| | Min-20% Prob | 11.20 | 12.98 |
| | Min-30% Prob | 9.92 | 10.89 |
| CausalNet | Min-10% Prob | 10.84 | 11.3 |
| | Min-20% Prob | 8.84 | 9.45 |
| | Min-30% Prob | 7.51 | 8.00 |
| C-E (*) | Min-10% Prob | 9.34 | 9.03 |
| | Min-20% Prob | 7.27 | 7.29 |
| | Min-30% Prob | 5.90 | 5.69 |
| C-H (*) | Min-10% Prob | 9.93 | 9.70 |
| | Min-20% Prob | 7.86 | 7.77 |
| | Min-30% Prob | 6.65 | 6.49 |

as a reference when performing causal reasoning tasks. Specifically, when querying LLMs a causal reasoning question, we first retrieve related general knowledge from a vector database constructed with the general knowledge dataset, through the pipeline of retrieval-augmented generation (RAG). Using the retrieved general knowledge as a basis, we employ the proposed goal-oriented prompt to steer LLMs to perform causal reasoning in a targeted manner, rather than aimlessly generating answers. This is the first step taken to advance LLMs towards *level*-2 causal reasoning.

# 6 Experiments

In this section, we will discuss two points: 1) the construction of the CausalProbe 2024 benchmark and its superiority; 2) the main implementation details ; 3) the result analysis of $G^2$-Reasoner and baseline methods. The CausalProbe 2024 benchmark and the source codes are presented in this URL: https://github.com/Haoang97/CausalProbe-2024.

## 6.1 Construction of CausalProbe 2024 and its superiority

We will introduce the construction procedure of the CausalProbe 2024 benchmark and provide a brief analysis. In general, the procedure contains two stages: 1) collecting the latest web articles to form a corpora; 2) employing an LLM to generate the question-answer pairs from the corpora.

**Corpora collection.** To ensure the quality of the collected corpora, we choose two well-known media: British Broadcasting Corporation (BBC) and the Guardian.[9] BBC [Link] is a British public service broadcaster founded in 1922, one of the oldest and largest broadcasting companies in the world. The Guardian [Link] is a national newspaper in the UK, established in 1821, with its online edition being particularly influential. We downloaded the latest articles from BBC and the Guardian. Specifically, the downloaded articles are from **January 1, 2024, to April 29, 2024**, later than the releases of the studied LLMs (Figure 1). The articles cover categories such as technology, environment, business, health, world news, culture, and climate, statistics of the downloaded articles are shown in Table 10.

---

[9]**NOTICE**: According to the licenses from the BBC and The Guardian, CausalProbe 2024 can only be used for **non-profit purposes**. All rights to the corpora used in CausalProbe 2024, including copyrights, are owned by the BBC and The Guardian.

**Bechmark construction.** We generated two sub-datasets from the same corpora using two different strategies. Since it's expensive to create them manually, we employ GPT 3.5 turbo as an assistant to automatically generate them. The overall pipeline diagram is shown in Figure 8.

- **CausalProbe-E**. We construct this dataset following the format of the CausalQA [6]. In detail, we first query the GPT 3.5 turbo to analyze an example data in CausalQA, then we offer GPT 3.5 turbo articles and ask it to generate Q&A pairs similar to the example data. Finally, the highest-quality Q&A pairs are selected as the reference, which is adopted in the prompt template of generating Q&A data. The related prompt templates are shown in Figure 14a, 14d.

- **CausalProbe-H**. We construct CausalProbe-H with another strategy. Specifically, first, we provide GPT 3.5 turbo an example article and ask it to summarize this article; second, we ask it to extract several cause-effect pairs based on the summary; third, we ask it to create some incorrect cause-effect pairs regarding this summary; finally, multiple multi-choice questions are generated according to these correct and incorrect cause-effect pairs. The highest-quality multi-choice Q&A data is selected as the reference, which is adopted in the prompt template of generating Q&A data. The related prompt templates are shown in Figure 14. The made-up fake cause-effect pairs can be used to examine the LLMs' genuine causal reasoning capability when encountering counterfactual disturbance term.

- **CausalProbe-M**. We construct an uncertain multiple-choice version, with the number of correct answers for each question ranging from 1 to 4. LLMs are required to distinguish the correctness or incorrectness of each causal proposition, avoiding LLMs relying on random guess to get correct answers. We provided GPT-4o mini[10] the prompt templates of CausalProbe-H and additional prompt to realize varying number of answers, shown in Figure 14. To ensure each question really has at least one correct answer, we manually checked the correctness of the provided answers.

Due to the broad range of topics in the corpus, it may contain ethically questionable content such as conflicts. Therefore, we used LLMs to filter the unethical questions from the preliminary generated data. Moreover, to comply with the context length limit of GPT 3.5 turbo, we removed articles exceeding 15,000 characters. After filtering, our dataset contains 3,461 unique Q&A data. We also provide statistical analysis and more details of CausalProbe 2024 in Appendix I.

**Superiority of CausalProbe 2024.** **1) Contextual information.** As we have simply mentioned, earlier three causal Q&A benchmarks lack the necessary background knowledge for each question. Although these benchmarks greatly promote the development of LLMs' causal reasoning, their formats are a little bit unreasonable. Even for humans, when we perform reading comprehension tests, we also need full context as a reference. Although we find that many of these questions can be answered by us, this heavily relies on our knowledge reserve, which is similar to LLMs. The knowledge reserve contributes their *level*-1 causal reasoning capability. Thus, we involve the contexts into CausalProbe 2024 as briefly as possible, which is more realistic, and they significantly improves the performance (Figure 5 and 6). **2) Hierarchical capability assessment.** Three datasets of CausalProbe 2024 hierarchically assess causal reasoning ability levels. CausalProbe-E assesses the genuine causal reasoning ability on novel problems. Furthermore, CausalProbe-H assesses the ability to identify misleading or deceptive causal propositions in new tasks. Last, CausalProbe-M assesses the ability to identify multiple valid causal statements in novel tasks, greatly ensuring that LLMa cannot obtain right answers through random guessing. By evaluating LLMs on these three datasets in sequence, we can determine their actual level of causal reasoning ability.

**Guarantee the freshness of CausalProbe 2024.** To doubly guarantee the freshness of CausalProbe 2024 than earlier benchmarks, we use an LLM's training data detection method, Min-$K$% Prob [53], to evaluate them as mentioned before. The detailed introduction for this method is in Appendix E. As the APIs of GPT and Claude no longer provide the logarithmic likelihood input texts, we only make this evaluation on open-source LLMs. The results in Table 3 show that CausalProbe 2024 is fresher. Here $K$ denotes the tokens in the bottom $K$ percent of log-likelihood.

We employ several measures to control the data quality of CausalProbe 2024. The detailed information is shown in Appendix H.

---

[10]During the contrustion of CausalProbe-M, OpenAI released the GPT-4o series models, which were more powerful and cheaper than GPT-3.5 turbo, so we adopted GPT-4o mini.

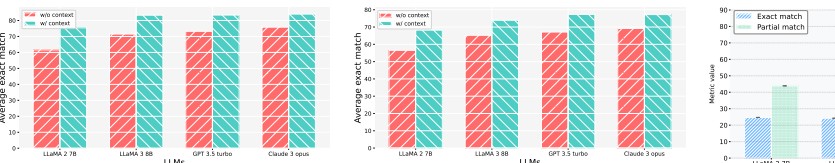

Figure 5: CausalProbe-E w/ and w/o contexts

Figure 6: CausalProbe-H w/ and w/o contexts

Figure 7: Result comparison between exact match and partial match

## 6.2 Implementation Details

All the experiments are conducted on the Ubuntu 20.04 system and NVIDIA RTX A6000 GPUs. For closed-source LLMs, i.e., GPT 3.5 turbo and Claude 3 opus, we call the API provided by their officials. We provide more implementation details in Appendix C.

## 6.3 Result Analysis

To evaluate $G^2$-Reasoner, we compare it with three common LLM reasoning methods, i.e., vanilla, chain-of-thought (CoT) [63], and retrieval-augmented generation (RAG) [32]. The "vanilla" refers to directly perform zero-shot inference. We evaluate four LLMs on CausalProbe 2024 and other three causal Q&A benchmarks, whose detailed introduction is presented in Appendix F. The full results are shown in Table 2. First, the LLMs' performance decreases monotonically as the benchmark corpora become more fresh. A counterintuitive phenomenon is that CoT usually perform a little worse than vanilla on three earlier benchmarks. This may indicate that CoT cannot improve reasoning on simple or common tasks for LLMs. For $G^2$-Reasoner, it mostly can outperform the vanilla method. However, limited by the scale of the general knowledge base and the power of vector databases, $G^2$-Reasoner cannot significantly outperform the baselines, which is the direction of our future efforts. In fact, RAG is just the ablated $G^2$-Reasoner, and it usually cannot reach the vanilla method, indicating the effectiveness of our goal-oriented prompt.

We evaluate the four LLMs on CausalProbe-M, where they showed a more significant performance decline compared to their results on CausalProbe-E and -H. Under exact match, which required all correct answers to be precisely identified, all models struggled. However, when using partial match, where missing some correct options was acceptable but selecting incorrect options was not, GPT and Claude performed relatively well, achieving accuracy rates of approximately 75% and 85% respectively. While these results demonstrate that current LLMs cannot fully comprehend each causal proposition, revealing limitations in their causal reasoning abilities, there is an encouraging finding: the models rarely make false positive errors when identifying causal relationships.

Note that $G^2$-Reasoner's performance relies on general knowledge bases. The reported results were obtained using a very small knowledge base (around 16 MB), yet $G^2$-Reasoner generally achieved non-marginal improvements. If we use a significantly larger one, such as Wikipedia API, performance can be boosted a lot. However, due to resource constraints, we couldn't repeat all experiments with it.

## 7 Conclusion and Future Outlook

This work investigates the causal reasoning capabilities of LLMs and argues that current LLMs are limited to *level*-1 causal reasoning. To verify this hypothesis, we introduce a new causal Q&A benchmark, CausalProbe 2024, revealing that LLMs struggle with causal reasoning in unseen contexts and primarily rely on the causal knowledge in training data . To fill this gap, we proposes $G^2$-Reasoner, a method that incorporates general knowledge and goal-oriented prompts into causal reasoning of LLMs. Experiments demonstrate that $G^2$-Reasoner can enhance the causal reasoning capability, particularly in fresh and counterfactual contexts. This work provides valuable insights into the current state of LLMs' causal reasoning and offers a promising attempt to move towards *level*-2 causal reasoning, bringing LLMs closer to reaching genuine causal reasoning capabilities. While this work takes an important step forward, it still does not enable LLMs to achieve *level*-2 causal reasoning. Further research in this field could potentially lead to significant advancements towards stronger artificial intelligence.

## 8 Acknowledgements

This work was partially supported by the National Natural Science Foundation of China (Grant No. 62372459, No. 62376282, No. 91948303). We express our sincere gratitude to Dr. Jie Tan for her valuable discussions and support from the National Natural Science Foundation of China (Grant No. 62402499).

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

# Supplement to "Unveiling Causal Reasoning in Large Language Models: Reality or Mirage?"

## Content

## A  Limitations

In this section, we discuss the limitations of our work, which include two parts. The first limitation is that the proposed causal reasoning method for LLMs is only a step forward, but cannot realize the genuine causal reasoning. On the other hand, enabling LLMs to perform causal reasoning like humans is challenging and requires further research. The second limitation is that we can not fully confirm the CausalProbe 2024 is unseen for LLMs. Specifically, the contents of CausalProbe 2024 are still part of human knowledge, and LLMs may have seen comparable information. For example, LLMs may not know one latest event, but their pre-training data may include a similar event. Thus, fully excluding the pre-training data from datasets is challenging, and we already ensure the CausalProbe 2024 is unseen for LLMs to a large extent.

## B  Broader Impacts

In this section, we discuss both the potential positive societal impacts and negative societal impacts of our work. The positive societal impacts include that our work facilitates understanding LLMs' current causal reasoning abilities. Besides, our work is a step forward for improving LLMs' causal reasoning. The potential negative societal include that our work cannot promote LLMs' causal reasoning abilities to the human level, thus, LLMs' causal reasoning results may still have errors.

## C  Full Implementation Details

All the experiments are conducted on the Ubuntu 20.04 system and NVIDIA RTX A6000 GPUs. For closed-source LLMs, i.e., GPT 3.5 turbo and Claude 3 opus, we call the API provided by their officials.

**Hyper-parameters.**    For LLM inference, we set the temperature parameter as $1.0$ for closed-source LLMs all the time, and set it as $0$ for open-source LLMs all the time. For CoT reasoning, we set the maximal new tokens as $128$, and we set it as $50$ for all other cases.

**Benchmark.**    All the used benchmarks are the .json or .jsonl format. If their original versions are not such formats, we convert them to .json or .jsonl format. For CausalNet, there exist a few missing values, and we have manually remove the invalid data. The filtered version of CausalNet will be uploaded in our anonymous link.

**$G^2$-Reasoner.**    $G^2$-Reasoner uses the RAG system to involve external general knowledge. We use the *Faiss* package [13] to construct the vector database. We use the Meta's *Contriever* [24] as the information retriever to perform vector database retrieval. For the construction of RAG system, we refer to the realization of self-RAG [3]. For the external general knowledge base, we use the answers of a general knowledge dataset (`https://huggingface.co/datasets/MuskumPillerum/General-Knowledge`) to form a document. We use *Contriever* and *Faiss* to embed and retrieve this general knowledge base.

**Training data detection.**    We follow the official repository of Min-$K$ Prob to perform detection, without any modification. However, as the Text-DaVinci-003 seems to be deprecated and current GPT 3.5 and GPT 4 are no longer support returning the input's log likelihood, we have to only perform this detection on open-source LLMs. We will continue to explore how to perform on OpenAI's alternative.

## D  Related Work

In this section, we review the related works in detail, including LLMs' reasoning, LLMs' causal reasoning, and LLMs' causal reasoning benchmarks.

### D.1  Reasoning in Large Language Models

Reasoning ability is crucial to LLMs' performance on tasks such as theorem proving, problem-solving, and robotics [41, 58]. Increasing interest in strong artificial intelligence has triggered debates on whether LLMs master reasoning abilities. [58] propose a comprehensive review of the foundation models' reasoning, they discuss reasoning tasks including commonsense reasoning, mathematical reasoning, logical reasoning, and causal reasoning. Commonsense reasoning refers to the reasoning process that utilizes commonsense and daily life experiences [9]. Rather than specialized knowledge, commonsense reasoning relies more on universally accepted knowledge. Commonsense question answering is an important method of commonsense reasoning test. Several commonsense question-answering datasets have been proposed to test LLMs' commonsense reasoning ability [19, 29, 74, 75]. Different from commonsense reasoning, mathematical reasoning requires the ability to master symbolic forms and formal definitions. Automated theorem proving is a commonly used task to test LLMs' mathematical reasoning ability [58]. [55] propose a framework for LLMs' theorem proving, which helps humans to prove theorems. LeanDojo [68] is a mathematical reasoning benchmark

consisting of theorems and proofs. Logical reasoning refers the process of using formal logic principles and rules to make deductions, draw conclusions, and solve problems [36, 58]. It involves applying logical rules and syllogistic reasoning to evaluate the validity of arguments and propositions. [62] propose an automatic evaluation framework for LLMs' logical reasoning ability, and find LLMs have difficulty in performing logical reasoning well. [59] shows that LLMs can learn from mistakes in logical reasoning. Causal reasoning is the process of identifying and understanding the cause-and-effect relationships between variables or events, which involves identifying potential causes and effects within a system or context [58]. One distinctive feature of causal reasoning is that it involves counterfactual reasoning, which means reasoning within a hypothetical scenario. [33] inspect LLMs' ability to distinguish between hypothetical and real-world scenarios. Our work focuses on the LLMs' causal reasoning, particularly counterfactual reasoning.

## D.2 Causal Reasoning in Large Language Models

Causal Reasoning tasks include causal discovery, cause attribution, and causal effect estimation [30, 58]. Causal discovery aims to recover the latent causal structure of variables, which could be a structural causal model (SCM). Cause attribution refers to uncovering potential causes behind a process, while causal effect estimation aims to investigate the effect of cause variables [26, 76]. While LLMs already show the ability to uncover causal relationships from context, their abilities have some limitations [37]. [4] suggests that LLMs are struggling to perform causal discovery tasks in hypothetical scenarios. [39] find GPT-3 has limitations in uncovering the causal structure among medical context. [77] illustrate the boundaries of LLMs' performance on causal discovery. Moreover, [38] indicates that LLMs' expertise may contain errors, which could undermine causal reasoning. Therefore, it's essential to incorporate external expert knowledge into LLMs to enhance LLMs' causal reasoning abilities. Counterfactual reasoning is an essential task of causal reasoning. LLMs with high counterfactual reasoning abilities have the strength to make predictions under various circumstances. The difference between counterfactual reasoning and other causal tasks is counterfactual reasoning involves reasoning in hypothetical scenarios [28, 58]. For example, "If cats were vegetarians, what results would happen?" is a counterfactual reasoning question [33]. Several studies conclude that LLMs have limitations when encountering hypothetical scenarios in counterfactual reasoning [33, 66].

## D.3 Causal Reasoning Benchmarks in Large Language Models

There have been extensive studies on causal reasoning benchmarks. Existing causal reasoning benchmarks are mainly causal question-answering datasets. CausalQA [6] is a dataset containing 1.1 million causal questions and answers. [6] employs language rules to extract causal questions from ten large question-answering datasets to form the CausalQA. CRAB [50] is a dataset that aims to assess LLMs' abilities to understand causal relationships among real-world events. [50] extract real-world stories during the past ten years and create CRAB based on these real-world stories. FCR [69] is a human-labeled dataset that includes 24K question-answering pairs. [69] collect data from Yahoo and employ crowd workers to generate the causal questions. However, the data they collected are between December 2020 and July 2021, which may be included in the pre-training corpora of the latest LLMs (e.g. LLaMA 3). Cladder [57] is a dataset that involves symbolic questions and corresponding ground truth answers, [57] employs causal graphs and structural causal models to generate the dataset. CausalProbe 2024 is different from the above benchmarks, as its contents are based on the latest and authoritative information, which is unlikely to be encompassed by the pre-training corpora of LLMs.

# E LLMs' Training Data Detection: Membership Inference Attack

In our work, we employ a pre-training data detection method [53] to confirm further that LLMs' pre-training data do not include our datasets to a large extent. MIN-K % PROB [53] is a pre-training data detection method. This method is based on a simple assumption: "When encountering an unseen example, the large language model (LLM) is likely to find a few words with low probabilities, while words in a familiar example are less likely to be assigned such low probabilities." Let $x = x_1, x_2, \ldots, x_n$ be a token sequence, and for one token $x_i \in x$, the log-like probability given its preceding tokens can be calculated as:

$$\log p(x_i|x_1, x_2, \ldots, x_{i-1}).$$

Then, each token's log-like probability can be computed. Let MIN-K%$(x)$ represent the set composed of the %K tokens with the lowest log probabilities among all tokens in sequence $x$. The average log-like probability of this set is:

$$\text{Min-}K\%\,\text{Prob}(x) = \frac{1}{N} \sum_{x_i \in \text{MIN-K\%}(x)} \log p(x_i|x_1, x_2, \ldots, x_{i-1}),$$

where $N$ is the number of elements in MIN-K%$(x)$. In the end, conditions on a threshold $\epsilon$, if Min-$K\%$ Prob$(x) > \epsilon$, then this sequence of text $x$ is determined not in pre-training data, otherwise $x$ is determined in pre-training data. One advantage of this method is that it does not need any information about the pre-training data nor does it need to train a reference model in advance.

## F  A Detailed Introduction of Used Benchmarks

### F.1  CausalNet

The CausalNet [4] is an LLM-generated dataset designed to support studies in causal reasoning and counterfactual analysis. With 1000 meticulously selected scenarios, CausalNet offers a wide range of causal and counterfactual inquiries, helping researchers delve into the complexities of cause-and-effect dynamics across different contexts. Each question-answering pair in the CausalNet contains a background context, a causal-effect question, or a counterfactual question, and the choices and answers of the question. However, compared with CausalProbe 2024, the background context of the CausalNet is limited, which may not provide sufficient basic information for LLMs to answer the questions better.

### F.2  COPA

The Choice Of Plausible Alternatives (COPA) evaluation [48] is a dataset including 1000 causal questions. The COPA aims to test LLMs' commonsense causal reasoning abilities. [48] employs an authoring methodology to secure broad topics, correct language, and high consensus among human evaluators. To be specific, they utilize several regulations to secure language clarity. Besides, they choose two different sources of questions that are confirmed broad to ensure the breadth. However, the articles they selected are mainly from August and September of 2008, which may be encompassed by LLMs' pre-training corpus.

### F.3  e-CARE

The CAusal REasoning dataset (e-CARE) [14] is a crowding-sourcing-based dataset, which contains 21K multi-choice causal reasoning questions. This dataset not only tests LLMs' ability to choose the correct causal statements but also examines the LLMs' ability to explain their choice. The e-CARE includes 21324 question-answering pairs and 13048 corresponding explanations. [14] collect statements of world knowledge and ask crowd workers to generate the causal facts according to the instructions. Then, the causal questions are generated by crowd workers based on these causal facts. However, LLMs' pre-training corpus may contain the collected statements about world knowledge.

## G  Additional Results

### G.1  Standard Deviations of Main Results

Due to the space limitation, we present the standard deviations of the results in Table 2 here. For two closed-source LLMs, i.e., GPT 3.5 turbo and Claude 3 opus, we find that setting the temperature value to 1.0 can achieve slightly better results than 0. We repeat these experiments three times, and the average results together with their standard deviations are shown in Table 6. The standard deviations are generally small.

## H  Quality Control

We have employed several measures to control the quality of CausalProbe 2024.

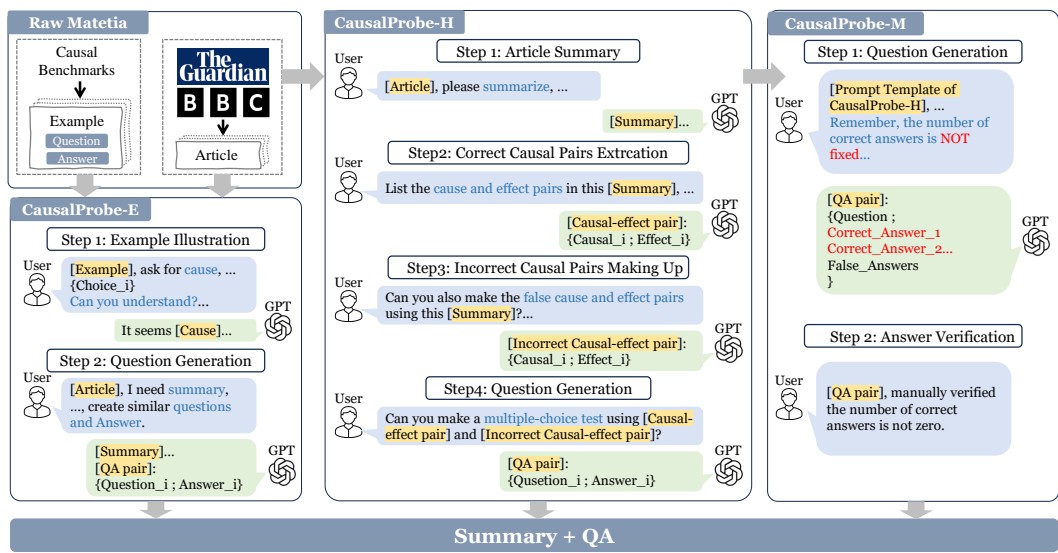

Figure 8: The pipeline of constructing CausalProbe 2024. CausalProbe 2024 consists of three datasets: CausalProbe-E, CausalProbe-H, and CausalProbe-M. They follow different strategies. In particular, CausalProbe-H introduces made-up fake cause-effect pairs, and can be used to examine the LLMs' genuine causal reasoning capability when encountering counterfactual disturbance term. CausalProbe-M features questions with varying numbers of correct options, preventing LLMs from providing right responses through random guessing.

- **Preparation**: Our corpus is sourced from two famous media with high quality. We further performed an initial cleaning of the corpus using regular expressions. Subsequently, we used the Google DLP API to detect sensitive information (such as pornography, violence, advertisements, etc.) in the corpus and removed any violating content.

- **Production**: We use GPT 3.5 Turbo, one of the best LLMs at the time, to construct the benchmark from the prepared corpus. To improve its quality, we experimented with different prompt templates and adopted the best.

- **Verification**: We write Python scripts to exclude incomplete/garbled items. Then we re-organize them to Json format for ease of reading. Finally we went through all items manually to find out problematic ones and excluded them.

We have made additional efforts to enhance the quality following a crowdsourcing pipeline. We recruited 17 volunteers, all of whom hold a master's degree or higher and are currently engaged in frontline research. Additionally, all volunteers are fluent in English. In the preliminary tests, we randomly sampled 20 questions from CausalProbe-H and asked each volunteer to answer them. We then recorded each volunteer's answer and their perceived difficulty level about this task (on a scale of 1-10, with 10 being the most difficult). The selection criteria required the perceived difficulty level to be no more than 7 and the accuracy rate to be no less than 80%. Ultimately, 13 out of the 17 volunteers met these criteria and were selected as qualified, and the test results are shown in Table 4.

Given the limited time and human resource, we performed quality control on a subset of CausalProbe 2024. Specifically, we randomly sampled 260 questions from CausalProbe-H and assigned each question to 3 volunteers randomly, using the Algorithm 1. Each volunteer received a total of 60 questions. After receiving their feedback, we treated those questions correctly answered by no less than two volunteers as high-quality data. Finally, 232 out of 260 questions were filtered out (temporarily called CausalProbe-HQ), achieving the qualification rate of 89.2%. Next, we use CausalProbe-HQ to evaluate four LLMs used in our paper again, whose results are shown in Table 5. The results show that all four LLMs still perform poorly on this subset, suggesting that their failure is primarily due to limited causal reasoning abilities rather than the errors in the benchmark. We will continuously perform quality control on all data through the aforementioned crowdsourcing pipeline and update the quality-controlled data on our GitHub repository.

**Algorithm 1** Question assignment algorithm

---

**Require:** List of all question IDs $Q$, List of volunteers $P$
 1: Shuffle $Q$ randomly
 2: Create $S[p]$ for each volunteer $p \in P$ to store their assigned questions
 3: **for all** question $q \in Q$ **do**
 4:     $P_{eligible} \leftarrow \{p \in P \mid |S[p]| < 60\}$
 5:     **while** $|P_{eligible}| < 3$ **do**
 6:         Shuffle $P$ randomly
 7:         $P_{eligible} \leftarrow \{p \in P \mid |S[p]| < 60\}$
 8:     **end while**
 9:     Assign question $q$ to the first 3 volunteers in $P_{eligible}$
10:     **for** $i = 1$ to 3 **do**
11:         $S[P_{eligible}[i]] \leftarrow S[P_{eligible}[i]] \cup \{q\}$
12:     **end for**
13: **end for**

---

Table 4: Statistical results of volunteer selection.

| ID | 1 | 2 | 3 | 4 | 5 | 6 | 7 | 8 | 9 | 10 | 11 | 12 | 13 | 14 | 15 | 16 | 17 |
|---|---|---|---|---|---|---|---|---|---|---|---|---|---|---|---|---|---|
| Diff level | 5 | 6 | 4 | 5 | 8 | 6 | 5 | 5 | 4 | 6 | 5 | 7 | 4 | 5 | 5 | 4 | 6 |
| Acc (%) | 80 | 75 | 90 | 100 | 80 | 90 | 70 | 100 | 85 | 90 | 70 | 95 | 90 | 85 | 95 | 100 | 85 |
| Qualified | ✓ | ✗ | ✓ | ✓ | ✗ | ✓ | ✗ | ✓ | ✓ | ✓ | ✗ | ✓ | ✓ | ✓ | ✓ | ✓ | ✓ |

Table 5: Results on a subset of high-quality data.

| Dataset | LLaMA 2 | LLaMA 3 | GPT | Claude |
|---|---|---|---|---|
| CausalProbe-H | 0.565 | 0.652 | 0.671 | 0.692 |
| CausalProbe-HQ | 0.547 | 0.660 | 0.698 | 0.733 |

# I Additional Introduction and Analysis of CausalProbe 2024

## I.1 Statistic of CausalProbe 2024

CausalProbe 2024 consists of 6,922 Q&A data, including two sub-datasets, each containing 3,461 Q&A data. Each sub-dataset covers topics such as technology, culture, business, climate, and world news. The statistic of CausalProbe 2024 in terms of the topics is shown in Figure 11. Moreover, all questions can be categorized into 'asking for cause' and 'asking for effect,' thus examining the LLMs' ability to reason about causes and effects in questions, respectively.

For CausalProbe-M, we visualize the distribution of the number of correct options in its Q&A data, shown in Figure 12. We find that two and three correct options occupy the majority and the distribution is simialr to be Gaussian distribution. We also visualize the distribution of query types (i.e., asking for cause or effect) in Figure 13.

## I.2 Addition Analysis

In this subsection, we discuss how the CausalProbe dataset tests the causal reasoning ability of LLMs. As discussed in Section 6.1, we collect the latest and authoritative web articles and utilize them to generate the dataset. To be specific, the downloaded articles are from **January 1, 2024, to April 29, 2024, later than the releases of many LLMs.** Consequently, these latest articles are unlikely to be directly incorporated into the pre-training corpus of some open-source LLMs (e.g. LLaMA 2 and LLaMA). Afterward, the questions generated by these articles are probably not included in the pre-training corpus of many LLMs. Therefore, LLMs are less likely to provide correct answers using their pre-training knowledge directly, which forces them to utilize their causal reasoning ability. Moreover, the questions cover a wide range of categories and encompass many real-world events that are highly discussable, which tests the causal reasoning ability of LLMs from multiple perspectives.

Table 6: Results with standard deviations of the studied LLMs on four causal Q&A benchmarks. The metric is exact match (EM). "Vanilla" denotes doing inference directly. "C-E", "C-H" represnt CausalProbe-E and CausalProbe-H. For GPT 3.5 turbo and Claude 3 opus, the temperature value is set to 1.0.

| | | LLaMA 2 | LLaMA 3 | GPT 3.5 turbo | Claude 3 opus |
|---|---|---|---|---|---|
| COPA | Vanilla | 0.752 | 0.937 | 0.943±0.002 | 0.992±0.002 |
| | CoT | 0.812 | 0.944 | 0.951±0.001 | 0.991±0.000 |
| | RAG | 0.757 | 0.912 | 0.939±0.005 | 0.988±0.003 |
| | $G^2$-Reasoner | 0.813 | 0.948 | 0.953±0.002 | 0.993±0.001 |
| e-CARE | Vanilla | 0.684 | 0.778 | 0.811±0.003 | 0.857±0.005 |
| | CoT | 0.697 | 0.770 | 0.807±0.006 | 0.863±0.002 |
| | RAG | 0.687 | 0.760 | 0.806±0.004 | 0.842±0.005 |
| | $G^2$-Reasoner | 0.701 | 0.779 | 0.821±0.001 | 0.854±0.003 |
| CausalNet | Vanilla | 0.673 | 0.857 | 0.895±0.003 | 0.930±0.002 |
| | CoT | 0.666 | 0.767 | 0.872±0.003 | 0.907±0.003 |
| | RAG | 0.650 | 0.860 | 0.894±0.007 | 0.910±0.004 |
| | $G^2$-Reasoner | 0.681 | 0.855 | 0.904±0.005 | 0.929±0.002 |
| C-E (*) | Vanilla | 0.616 | 0.715 | 0.730±0.002 | 0.755±0.002 |
| | CoT | 0.636 | 0.720 | 0.740±0.002 | 0.751±0.001 |
| | RAG | 0.621 | 0.704 | 0.743±0.001 | 0.756±0.001 |
| | $G^2$-Reasoner | 0.642 | 0.718 | 0.747±0.003 | 0.762±0.002 |
| C-H (*) | Vanilla | 0.565 | 0.652 | 0.670±0.001 | 0.688±0.003 |
| | CoT | 0.573 | 0.644 | 0.662±0.004 | 0.700±0.002 |
| | RAG | 0.575 | 0.655 | 0.678±0.001 | 0.687±0.001 |
| | $G^2$-Reasoner | 0.582 | 0.658 | 0.690±0.003 | 0.701±0.004 |

## I.3  Potential bias

As known for us, any machine model has its own inductive bias [52]. Similarly, using GPT 3.5 turbo to build CausalProbe 2024 introduces potential biases. They mainly consist of the following types of bias:

- **Model bias**: GPT-3.5 turbo may inherit biases and limitations from its own training data.

- **Generation bias**: GPT-3.5 turbo may frequently produce certain types of questions or answers.

- **Language and cultural bias**: Both CausalProbe 2024's English news corpus and GPT-3.5 Turbo's predominantly English training data may introduce Western or Anglophone biases.

To mitigate these potential biases, we propose the following measures:

- **Diversification of data sources**: We can incorporate diverse corpus sources and more methods, such as manually created Q&A pairs or data generated by different LLMs.

- **Manual check**: We can conduct a thorough human review and curation for CausalProbe 2024's biases.

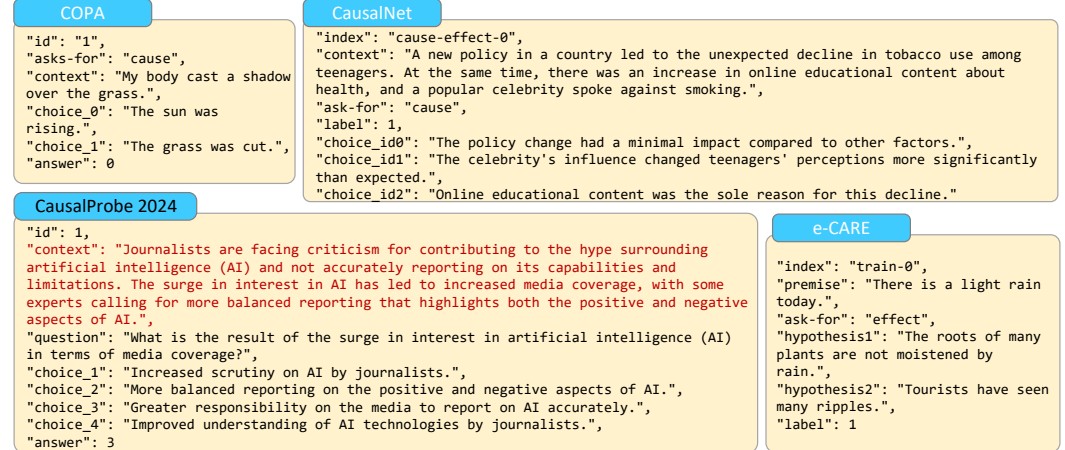

Figure 9: Examples of four Causal Q&A benchmarks used in this work. These four benchmarks come from different corpora, and exhibit the similar format and difficulty. In particular, CausalProbe 2024 additionally provides the context for each question, which describes the question's background knowledge. Note that other three datasets also have "context" or "premise", but actually, this is a part of the question itself, not the background knowledge.

Figure 10: Statistics of the downloaded articles in terms of their topics.

| Categories | BBC | The Guardian |
|---|---|---|
| Science | 220 | 251 |
| Culture | 381 | - |
| Climate | 121 | - |
| Technology | - | 340 |
| Health | - | 204 |
| Environment | - | 329 |
| Business | 101 | 352 |
| World News | 69 | 934 |
| Others | 75 | - |
| Total | 967 | 2702 |

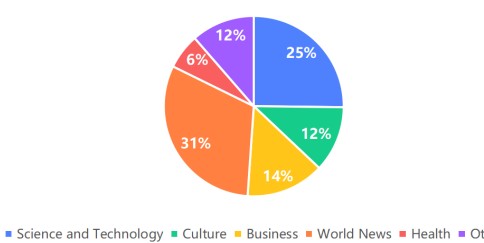

Figure 11: The statistic of the number of each topic's data in CausalProbe 2024.

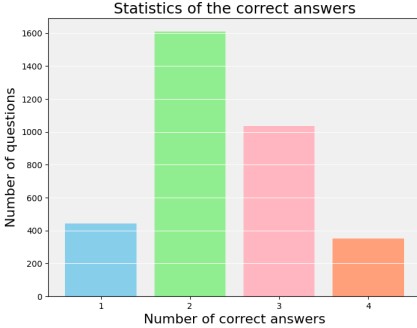

Figure 12: The statistic of the number of correct options in each Q&A data of CausalProbe-M.

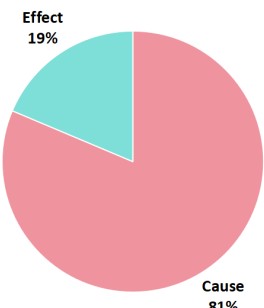

Figure 13: The statistic of query types in each Q&A data of CausalProbe-M.

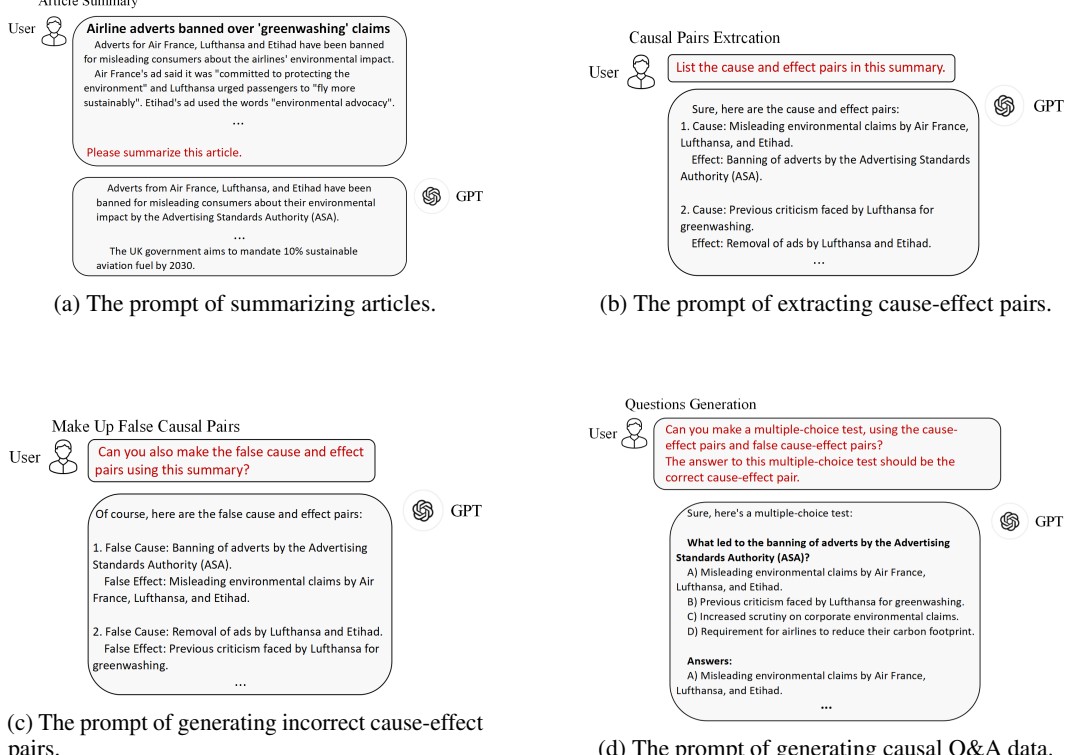

(a) The prompt of summarizing articles.

(b) The prompt of extracting cause-effect pairs.

(c) The prompt of generating incorrect cause-effect pairs.

(d) The prompt of generating causal Q&A data.

Figure 14: The prompt templates used in constructing CausalProbe 2024.

