# OpenReview forum: "Unveiling Causal Reasoning in Large Language Models: Reality or Mirage?"
_NeurIPS.cc/2024/Conference — NeurIPS 2024 poster_

### Official Review · Reviewer_ZgYU · 2024-07-09

**Soundness:** 3
**Presentation:** 3
**Contribution:** 2
**Rating:** 6
**Confidence:** 4

**Summary:**

The paper is concerned with the question of whether LLM can perform causal reasoning on a human-like level by incorporating contextual information in their decision and answer process. The authors argue that many everyday causal inferences are not purely logical but take into account general knowledge and intention. Therefore, a distinction between 'shallow' (level-1) and 'human-like' (level-2) causal reasoning is proposed. Two versions of a novel  "CausalProbe-2024" causal Q&A data set for benchmarking level-2 reasoning capabilities of LLM are presented. The presented benchmark is sourced from recent news sources. The authors, therefore, claim that the presented data is unlikely to be contained in the training set of the tested LLM (LLaMa 2/3, GPT-3.5 Turbo, and Claude 3).

To tackle level-2 causal reasoning, the authors propose a theory on how cause-effect pairs get instantiated into natural language sentences and further argue that a general world context affects the instantiated cause and effect nodes. To reason in this kind of setting, the authors propose a "$G^2$-Reasoner" to incorporate general knowledge and inherent goals via additional context information retrieved from a general knowledge base.

Evaluating multiple LLMs on the CausalProbe data set shows a strong deterioration in performance compared to earlier causal reasoning datasets. The proposed $G^2$-reasoner helps improve reasoning capabilities over a naive baseline approach and performs on par with Chain-of-Thought or retrieval-augmented generation approaches.

**Strengths:**

Overall, the author's proposal of assessing the causal reasoning capabilities of LLM in context is a valid contribution and improves in realism and difficulty over existing benchmarks. The work is well embedded into existing related work on the causal reasoning capabilities of LLMs. Relations to existing causal Q&A data sets are drawn, and previous examinations of LLM causal reasoning capabilities are discussed.

The novel data set is constructed by extracting information from recent news articles of two major news sources, covering events that lie after the information cut-off of all tested LLMs (after Jan 1st, 2024). To support the thesis of querying the LLMs on unseen information, LLama models are tested via the Min-K% Prob technique on whether the models might have memorized the collected information. Compared to older datasets, CausalProbe seems to be composed of more novel information than COPA, e-CARE, or CausalNet data sets.

The authors contribute a possible explanation of how causal information is instantiated in natural language via a high-level SCM. This involves a general context, drawing information from world knowledge, and incorporating the naturally occurring diversity of expressing such relations.

The experimental setup and individual steps of the $G^2$-reasoner are described clearly. Prompt templates are provided. The approach is compared to a 'vanilla' approach, Chain of Thought (CoT), and a retrieval-augmented generation (RAG), giving the LLMs access to a general knowledge base.

**Weaknesses:**

While I agree with the authors that benchmarking LLMs for causal relations in context is a more difficult task than pure logical causal reasoning, the paper remains vague on the particular effect that the additional context might impose in terms of the cause-effect pairs as discussed in Sec. 5 / Fig. 4.

(1) Generally, the described distinction between level-1 and -2 causal reasoning (def. 2 and 3) seems to be about how LLMs draw information from either memorization (information embedded in the model weights) or dynamically from the input text. LLMs are known to have difficulties when reasoning over non-memorized facts. In this regard, the general shortcomings of LLMs are non-surprising and have already been presented in previous works (e.g., Kiciman et al., 2024; Jin et al., 2024; Zecevic et al., 2023). In that regard, the proposed $G^2$-reasoner seems to feature no explicit mechanism to help improve on the causal reasoning of LLMs other than providing general background knowledge. No alternative prompt variations or combinations with, e.g. CoT or fine-tuning, are presented to improve results.


(2) My main concern is the quality of the collected data set itself. While the authors do clearly describe the automated process of extracting information from credible news sources, no human involvement in quality checking is mentioned. Given the more implicit nature of causal relations considered in this paper, many answers can not be directly inferred but need to be extrapolated from the texts (as intended by the authors). However, checking on the first ten entries of CausalProbe-H seems to reveal a generally poor quality of the samples. In detail, I get the impression that multiple answer options per sample can be valid. Given that only a single answer is indicated as 'correct' by the data set, I question the significance of the reported results. In detail:

* id 3: correct marked answer 2 is implicit, but so could be 3 and 4.
* id 4: 1, 2 and 4 are possible answers with partial contribution to the effect.
* id 5 - 1 and 2 are both correct.
* id 7 - 2 and 4 describe partial factors that could both be valid.

The main problem seems to stem from the unclear definition of what counts or does not count as a causal factor for a specific scenario. As a result, alternative answers --other than the one indicated by the dataset-- might contribute to the causal effect relation and, therefore, could be viewed as reasonable answer options. The authors might have either adopted an answer format capable of accepting multiple correct answers or might impose a more extensive quality checking to prevent the appearance of alternative valid answer options.


(3) The partition of samples into causalProbe-S and -H is unclear to me. The authors mention that "the highest-quality Q&A pairs are selected as the reference" (l.289) but do not describe under which criteria this selection is done.


(4) The authors mention in Appendix (C) "Full Implementation details" to have set the temperature to "1.0 for closed-source LLMs [...], and set it as 0 for open-source LLMs [...].". Non-zero temperature evaluations might introduce additional variance to the results. However, the authors do not quantify the possible variance in in their evaluation results.


Minor comments:

* Typo in the caption of table 3: "represnt".

* While I agree in general with the discussion in Sec. 4.1 about causal event sequence and sequence of appearance, it dismisses possible generalization capabilities of LLMs. Claims 1) and 2) are provided without reference or experimental evidence. The authors might want to tune down these claims.

* I would advise not captioning Fig. 2 (b) as a "sequential causal relationship" but rather as a "sequence of appearance" since --as the authors mention correctly-- word sequences do not need to imply any causal relations inherently.

* Typo "we proposes G2-Reasoner" (l.340)

**Questions:**

My questions mainly concern the weaknesses mentioned above. In detail, I would like the authors to comment on the following:

I) Regarding (2): could the authors clarify the measures that were imposed to ensure the quality of the collected data samples? How do the authors view the possibility of multiple valid answers with regard to the proposed data set format and evaluation results?


II) Regarding (1) and (2), I would like to ask the authors to clarify the expected effects of the context on the causal relation. In detail, could the authors comment (e.g., in the context of epistemic reasoning / modal logic) how 'reasonable/possible' and 'correct/necessary' answers would be distinguished in their setup and which implications would follow for their data set design?


III) Regarding (3): is the partition of the samples performed by human judgment or via some automated metric? Which criteria are used to judge sample difficulty?


IV) Regarding (4): Can the authors quantify the possible randomness-induced variance for the closed-source models?

**Limitations:**

In their paper, the authors correctly address the causal reasoning limitations of LLMs in general and their $G^2$-reasoner in detail.

The authors seem to have not performed or do not mention a human-backed review of the data set contents. The quality in terms of answer texts and evaluation results could be improved by establishing such a process.

The data set seems to be free of ethically questionable content; However, the applied filtering process was not disclosed.

---

> ### Author Rebuttal · Authors · 2024-08-07
>
> Thank you for constructive comments and they are valuable for improving our paper. In the following, I will address your concerns one-by-one.
>
> > Q1. The paper remains vague on the particular effect that the additional context might impose in terms of the cause-effect pairs as discussed in Sec. 5 / Fig. 4.
>
> A1. In my understanding, what you refer to as 'the additional context' means general knowledge. We model a textual causal reasoning task through a SCM in Fig. 4. General knowledge, i.e., variable $C$, **drives the causal relationship between two events**. Without it, the derived causal relationship may contradict the objective causal laws. Thus, general knowledge serves as a **guide** for LLMs' causal reasoning.
>
> > Q2. Weakness (1)
>
> A2. First, level-2 causal reasoning refers to the **genuine complex reasoning** akin to human cognition, rather than merely retrieving information from sources.
>
> Our paper's main contribution is **exploring the boundary of LLM causal reasoning abilities and identifying the causes of their limitation by CausalProbe 2024**. Previous works did not systematically studied this. We also proposed the G$^2$-Reasoner for LLM's **textual causal reasoning**. It involves general knowledge and a goal-oriented prompt: the former serves as a guide, and the latter steers LLMs to identify correct cause-effect pairs and reach the intended goal, alleviating off-target generation.
>
> CoT and finetuning are both effective for LLMs. While finetuning works well for specific tasks, it often leads to catastrophic forgetting, conflicting with our general-purpose goal. CoT is versatile and as a baseline in our paper. Incorporating CoT into G$^2$-Reasoner only boosts performance slightly, so we didn't combine it with G$^2$-Reasoner to avoid additional inference costs.
>
> > Q3. Weakness (2) & Question (I)
>
> A2. Upon review, we found that some questions might have more than one correct option if we only consider the "problem" itself. However, within the given "context", the unique answer provided by us is usually the most relevant option.
>
> First, we discuss the works we **have done** to ensure the quality. To further improve our benchmark, we have taken **two additional steps** based on your comment. **Please refer to the *Author Rebuttal* for more details.**
>
> > Q4. Weakness (3) & Question (III)
>
> A4. We clarify that CausalProbe-E and -H **are not created by partioning existing data** based on a difficulty metric. Instead, they are **constructed from scratch** using different strategies but the same corpus. We presented the methods for constructing them in **Section 6.1 and Figure 7**. I am glad to introduce their differences here.
>
> CausalProbe-E follows the format of CausalQA [1]. We provided GPT with original corpus and samples from CausalQA and asked GPT to generate Q&A data by imitating them.
>
> The construction of CausalProbe-H is more complex. We provided GPT with an article and asked it to generate several cause-effect pairs. These pairs were classified as true (accurate information) or false (distortions). GPT then generated Q&A pairs from these cause-effect pairs. Three co-authors rated the Q&A pairs for quality, selecting the best one. This top-quality Q&A pair, along with its source article, was used as an in-context example. We then used this in-context example with a new article to prompt GPT to generate new data. Unlike CausalProbe-E, CausalProbe-H tests LLMs' ability to reason and identify erroneous causal information.
>
> [1] Bondarenko et al. Causalqa: A benchmark for causal question answering. COLING 2022.
>
> > Q5. Weakness (4) & Question (IV)
>
> A5. During the experimental phase, we discovered that setting the temperature of the closed-source models to 1.0 slightly improved performance compared to 0, albeit with variance. Due to budget constraints, we did not initially repeat the experiment to obtain the standard deviation. Now, we have repeated the experiments **two more times**, and the  standard deviations are shown in Table 4 in PDF.
>
> > Q6. Minor comments
>
> A6. We have polished our paper again to correct the remaining typos and grammar issues.
>
> For Sec. 4.1 and the caption of Fig. 2 (b), we have modified them following your advice: 1) we have weakened our claims 1) and 2): "Based on the above discussion, there are two **possible** issues: 1) ..."; 2) we have changed "sequential causal relationship" to "sequence of appearance".
>
> > Q7. Question (II)
>
> A7. The context serves as a **reference** for specific causal reasoning tasks. Without constraints, causal or general reasoning can be aimless, especially for LLMs. Given only a question, LLMs might find multiple "reasonable" options. However, with the context provided, LLMs can more easily identify the "correct" option, which is the optimal answer. In our benchmark design, we provided context for each question, **distinct from previous causal Q&A benchmarks**. Figure 5 & 6 compare the performances w/ and w/o contexts, showing that context helps LLMs' causal reasoning.

---

> ### Comment · Reviewer_ZgYU · 2024-08-08
>
> Dear authors,
> I highly appreciate the extensive efforts that you put in place to improve the quality of your paper. I believe that, both, the human annotations as well as the newly added multi-choice options greatly contribute in consolidating the notion of causality within the data set. Although, the reported sample rejection rate found in the study seems to be quite low compared to my (admittedly very small) sample set, I believe that the procedure is sufficient to ensure a well curated collection of items.
>
> Apart from that, all my other questions other have been answered sufficiently. As such, I have raised my score to weak accept and recommend the acceptance of this paper.

---

> > ### Author Response · Authors · 2024-08-09
> >
> > Dear Reviewer,
> >
> > We sincerely appreciate your support for our paper and are particularly grateful for your invaluable comments. These comments have significantly enhanced the quality of our benchmark, making it more precise and reliable.
> >
> > We have carefully incorporated our responses and clarifications into the paper and are currently conducting thorough quality control on the remaining data in CausalProbe 2024.
> >
> > The opportunity to discuss our work with you has been immensely constructive, leading to substantial improvements in our research. Your time, expertise, and thoughtful feedback are deeply appreciated. Should you have any further suggestions or comments, please do not hesitate to share them with us.
> >
> > Thank you once again for your invaluable contribution to the enhancement of our research.
> >
> > Best regards,
> >
> > Authors of Paper 8470

---

### Official Review · Reviewer_NPEc · 2024-07-12

**Soundness:** 3
**Presentation:** 3
**Contribution:** 3
**Rating:** 6
**Confidence:** 3

**Summary:**

The authors proposed a new causal reasoning framework, to improve the causal reasoning capacity of LLMs, with inspiration drawn from causal graph theory and human reasoning process. The work utilized "general world knowledge" as a component to take a step closer to making LLMs perform a more human-like causal reasoning process. The authors also developed a new benchmark and carried out experiments evaluating how different LLMs perform on their benchmark.

**Strengths:**

The paper is well-written and easy to follow. The authors draw inspiration from the human causal reasoning process and provide an analysis of why LLMs cannot perform "genuine" causal reasoning, from not only a methodological perspective but also an empirical perspective.

**Weaknesses:**

1. The key hypothesis of this work is that LLM is capable of "level-1" reasoning, but lacks the capacity of "level-2" reasoning. Given that this work is centered around this hypothesis, a clearer definition and illustrative example of level-1 and level-2 should be provided.
2. I suggest the authors provide an example of a "random exogenous variable" in line 238. I can understand what this variable is from the causal graph framework, but it would be helpful to better clarify it in your setting.
3. Can you provide a more detailed explanation of how equation (1) plays a role in your G2 reasoner, and how can you compute this term?

**Questions:**

see weaknesses.

**Limitations:**

The author didn't provide a separate limitations section. One limitation the authors mentioned in the paper is that they only consider causal reasoning tasks with single cause-effect pair. Another limitation is that their approach doesn't enable LLMs to achieve level-2 causal reasoning, but rather provides insights into it.

---

> ### Author Rebuttal · Authors · 2024-08-07
>
> Thank you for constructive comments and they are valuable for improving our paper. In the following, I will address your concerns one-by-one.
>
> > Q1. The key hypothesis of this work is that LLM is capable of "level-1" reasoning, but lacks the capacity of "level-2" reasoning. Given that this work is centered around this hypothesis, a clearer definition and illustrative example of level-1 and level-2 should be provided.
>
> A1. Thank you for point out this ambiguity. Here we provide more clear definitions for two causal reasoning levels.
> - **Level-1** causal reasoning refers to retrieving existing causal knowledge embedded in model parameters and the given contexts. It is usually fast and suitable for handling simple causal relationships. Current LLMs are at this level.
> - **Level-2** causal reasoning mimics human cognition, allowing LLMs to use powerful reasoning mechanisms and existing knowledge to infer causal relationships, even with unfamiliar tasks. This process is slower but capable of handling unknown causal knowledge.
>
> Note that level-2 reasoning is not always better, as it is less efficient and cost-effective. Ideally, LLMs would adaptively choose the appropriate reasoning mode based on task difficulty. However, our paper shows that LLMs lack level-2 causal reasoning capabilities, preventing this adaptability.
>
> > Q2. I suggest the authors provide an example of a "random exogenous variable" in line 238. I can understand what this variable is from the causal graph framework, but it would be helpful to better clarify it in your setting.
>
> A2. Of course. The formula $h(X,Y,\epsilon)=T$ represents a natural language generation process that contains cause-effect information. Here, $X$ and $Y$ represent the concepts of cause and effect, respectively, and $T$ is the textual expression of this causal relationship through the mapping $h$. The variable $\epsilon$ represents **various factors** in generating readable text from the causal concepts $X$ and $Y$, **such as language type, context, and mode of expression (e.g., active or passive voice)**. While $\epsilon$ contributes to the **diversity and flexibility of natural language**, it also complicates LLM's causal reasoning from a linguistic perspective.
>
> For example, consider the concepts "smoking" ($X$) and "lung cancer" ($Y$). With different $\epsilon$, we can get different natural language expressions: 1) "A history of smoking is a common risk factor for lung cancer." 2) "Knowing that smoking greatly increases the risk of lung cancer, why take the risk?" Both sentences imply the same causal relationship but differ linguistically.
>
> > Q3. Can you provide a more detailed explanation of how equation (1) plays a role in your G2 reasoner, and how can you compute this term?
>
> A3. Certainly. We are happy to explain the role and motivation bebind Eq. (1). This equation represents the task of **inferring the most probable effect from a cause given a context**, in the language of statistics. To achieve this goal, we need the cause ($X$), the natural language expression of the causal proposition ($T$), and general knowledge ($C$). In our setup, $X$ and $T$ are known, so we need access to a complete general knowledge base ($C$). With this, LLMs can ideally reason out the correct effect, and mathematically, the total probability formula in Eq. (2) holds.
>
> However, we emphasize that what Eq. (1) and Eq. (2) provides is the **technical motivation** from a causal inference perspective, and we **do not need to calculate them explicitly**. We use such technical motivation to design the G$^2$-Reasoner, which integrates the general knowledge base into LLMs' causal reasoning processes through RAG.

---

> > ### Comment · Reviewer_NPEc · 2024-08-09
> >
> > Thank you for the response. I've read it carefully and my questions have been addressed. I will keep my current score.

---

> > > ### Author Response · Authors · 2024-08-10
> > >
> > > Dear Reviewer,
> > >
> > > We sincerely appreciate your support for our paper and are particularly grateful for your invaluable comments.
> > >
> > > The three comments you raised have greatly improved the clarity and readability of our paper. We have carefully revised our paper based on them,  especially regarding the understanding of SCM in the causal reasoning tasks of LLMs .
> > >
> > > The opportunity to discuss our work with you has been immensely constructive, leading to substantial improvements in our research. Your time, expertise, and thoughtful feedback are deeply appreciated. Should you have any further suggestions or comments, please do not hesitate to share them with us. Thank you once again for your invaluable contribution to the enhancement of our research.
> > >
> > > Best regards,
> > >
> > > Authors of Paper 8470

---

### Official Review · Reviewer_yorJ · 2024-07-12

**Soundness:** 3
**Presentation:** 3
**Contribution:** 3
**Rating:** 6
**Confidence:** 4

**Summary:**

The paper investigates the causal reasoning capabilities of LLMs and argues that current LLMs are limited to shallow (level-1) causal reasoning. To support this claim, the authors introduce a new benchmark, CausalProbe-2024, which reveals that LLMs struggle with causal reasoning in fresh and unseen contexts. To address this, the authors propose G2-Reasoner, method that incorporates general knowledge and goal-oriented prompts to enhance causal reasoning capabilities. Experiments show that G2-Reasoner improves performance, particularly in novel and counterfactual scenarios.

**Strengths:**

- The paper goes beyond simple retrieval from LLM memory causal reasoning tasks.

 - They attempt to establish the sensitivity of the model for pretraining

- The authors also propose a new benchmark CausalProbe-2024 to evaluate level-2 reasoning.

- The authors identify fundamental limitation in the current architecture of LLMs as authoregressive next token prediction mechanism (which is intuitive though) for causal reasoning.

**Weaknesses:**

- The long-term viability of this approach is uncertain. Continuous advancements in model architecture and training techniques might be required to truly enable level-2 reasoning, and the proposed method might only be a temporary solution. However I understand that this paper still makes a good progress.

- Unless I missed it, I did not find much detail on RAG since it is so knowledge-specific. It isnt always aplicable. Also the results dont show a big improvement for G2 reasoner. It then begs the question how dependent was the model the knowledge of RAG.

- paper primarily addresses simple, single cause-effect pairs which is a bit limiting

- the variances arent given

**Questions:**

Just a minute comment, for better readability, I would appreciate it if the authors could an identifier(eg bold) in the Tables 2,3 just to make it easier to read.

**Limitations:**

Yes

---

> ### Author Rebuttal · Authors · 2024-08-07
>
> Thank you for constructive comments and they are valuable for improving our paper. In the following, I will address your concerns one-by-one.
>
> > Q1. The long-term viability of this approach is uncertain. Continuous advancements in model architecture and training techniques might be required to truly enable level-2 reasoning, and the proposed method might only be a temporary solution. However I understand that this paper still makes a good progress.
>
> A1. We partially agree with this comment. In the future, more powerful model architectures, training methods, or new forms of AI may eventually achieve genuine causal reasoning even surpass humans. However, this may **take a long time**. Therefore, our current efforts in causal reasoning for LLMs are **meaningful at the current development stage**.
>
> > Q2. Unless I missed it, I did not find much detail on RAG since it is so knowledge-specific. It isnt always aplicable. Also the results dont show a big improvement for G2 reasoner. It then begs the question how dependent was the model the knowledge of RAG.
>
> A2. Thank you for this insightful comment. The implementation details of RAG was presented in Appendix C.
>
> In our paper, RAG incorporates general knowledge into LLM causal reasoning. As you noted, complete and high-quality knowledge bases are crucial for RAG performance. Our reported results for G$^2$-Reasoner are based on a **very small general knowledge base (about 16 MB), yet it achieved non-marginal performance improvement**. If we use a complete one, such as Wikipedia API, the performance can be boosted a lot. However, due to resource constraints, we couldn’t repeat all experiments with Wikipedia API. Instead, we are creating a more complete offline general knowledge base and will open source it, which is helpful for LLM causal reasoning and other fields.
>
> > Q3. Paper primarily addresses simple, single cause-effect pairs which is a bit limiting.
>
> A3. Thank you for your insightful comment. Our paper is **the first to comprehensively explore the boundary of LLMs' causal reasoning abilities**. We created a benchmark with over 6,000 multiple-choice questions. However, our benchmark **is not simply established on single cause-effect pairs**. To pick the correct option, especially for CausalProbe-H, LLMs are also required to identify incorrect but confusing cause-effect pairs. Our benchmark construction method was deliberately designed to achieve this goal, and the details were presented in Sec. 5 "Bechmark construction" and Figure 7. Following your suggestion, we will construct more diverse and complex causal reasoning tasks for LLMs in the future.
>
> > Q4. The variances arent given.
>
> A4. Of course, thank you for your suggestion! Due to budget constraints at the time, we ultimately did not repeat the experiment multiple times to obtain standard deviations. Currently, we have repeated the experiments of closed-source models (GPT 3.5 Turbo and Claude 3 opus) **anthor two times** and the results together with standard deviations are shown in Table 4 in PDF.
>
> > Q5. Just a minute comment, for better readability, I would appreciate it if the authors could an identifier(eg bold) in the Tables 2,3 just to make it easier to read.
>
> A5. Thank you for this reminder. We have highlighted the best results under each benchmark and each LLM in **bold** in Table 2 and Table 3. In addition, we have completely polished our paper again to make it easier to read.

---

> > ### Comment · Reviewer_yorJ · 2024-08-10
> > **Response**
> >
> > I would like to thank the authors for responding to my concerns. I have now raised the score to weak accept. I would appreciate it if the author would include variances in the final version.

---

> > > ### Author Response · Authors · 2024-08-11
> > >
> > > Dear reviewer,
> > >
> > > We sincerely appreciate your support for our paper and are particularly grateful for your invaluable comments.
> > >
> > > We have carefully repeated our experiments and included the standard deviation into our paper during the author response phase. This well demonstrates the stability of the performance of G$^2$-Reasoner and baseline methods when the temperature of LLMs is larger than 0.
> > >
> > > The opportunity to discuss our work with you has been immensely constructive, leading to substantial improvements in our research. Your time, expertise, and thoughtful feedback are deeply appreciated. Should you have any further suggestions or comments, please do not hesitate to share them with us. Thank you once again for your invaluable contribution to the enhancement of our research.
> > >
> > > Best regards,
> > >
> > > Authors of Paper 8470

---

### Official Review · Reviewer_ZNie · 2024-07-13

**Soundness:** 2
**Presentation:** 3
**Contribution:** 2
**Rating:** 5
**Confidence:** 3

**Summary:**

This paper studies the autoregressive mechanism of the transformer-based LLMs and their ability to reason causally and introduces a new Q&A-based benchmark named CausalProbe-2024 with around 3k Q&A pairs (believed to be unseen by the existing LLMs). The paper further introduces a reasoning strategy that considers general knowledge and the goal of performing a task called G2reasoner. The proposed G2-reasoner is based on a RAG based framework that encodes the information related to general knowledge in the form of a vector database, and the goal orientation is simulated by modifying the prompt that steers an LLM to be an intelligent causal reasoner and provides the correct causal relationship between the events and reason about them.

The paper reports the empirical evaluation results in over 3 widely used benchmarks (COPA, e-CARE, and CausalNet) along with the two versions of the proposed Causal Probe dataset.

**Strengths:**

* The paper raises an interesting question regarding the causal reasoning in LLMs, and the research goal established by the paper with the underlying question regarding causality is timely.

* The paper’s primary strength comes from creating a newly created benchmark for facilitating the study of causal reasoning in LLMs when compared to humans. Formulating a formal causal reasoning benchmark in a natural language is an extremely challenging task, and this work will take a small step toward making the causal reasoning evaluation effective.

* The paper highlights the training data detection using MinK% Prob in Table 3 for the existing causal benchmarks/datasets, which will be helpful for future research. With evaluation and comparison with the widely used causal reasoning benchmarks, the paper highlights some findings for the open-weight and popular proprietary models.

**Weaknesses:**

* The definitions of level 1 and level 2 are not concretely stated. Though lines 48 to 51 vaguely describe it, and the expanded version of the definitions is present in section 3, line 126 to line 135, it would be better to make them more concrete. In the current version, the definition is not very clear when it talks about complex or unknown tasks. A clear distinction between the levels would help the reader/research community to explore it further in future works.

* The results of the studied LLMs on four causal Q&A benchmarks, as shown in Table 2, highlight very little improvement of the G2-reasoner compared to the other approaches. Moreover, the motivation/inspiration taken for the G2 reasoner is generic and not causal specific (specifically the context and the RAG part) and may help improve the performance over tasks where causal reasoning is not required.

* Figure 5 and Figure 6 shows the performance improvements when provided with the context, which is generally true for all the LLM evaluations, and the observation may not be causal reasoning specific. It may be noted that the G2-reasoner approach may also be applicable to other non-causal tasks and may have a similar result in other non-causal benchmarks. The proposed strategy will be more reliable when there is a direct effect over the causal-reasoning benchmark.

**Questions:**

* The assumption behind the autoregressive models made in line 159, “in a sequence, the current value is determined by past values, not related to future values,” may not be entirely correct. Specifically, highlighting “the current value is not related to future values” may be incorrect.

* Line 148 raises a very good point regarding the complexity of natural language, and there can be many different ways or sentence patterns to express the same information. However, the evaluation metrics used in the paper is Exact match (Table 2, Figure-5, Figure-6).  It would be good to consider a better evaluation metric while performing the evaluations.

* The claims made in the para [line 52 to line 61] and figure 2 are not completely justified in paras [line 156 to line 180], concluding on the statement [lines 178 to 180] “Thus, the autoregression mechanism makes LLMs’ causal reasoning primarily rely on correct causal knowledge in a large number of training corpora, i.e., the level-1 causal reasoning”. It would be great if the authors could provide a more detailed justification and talk about level 2 as well. Currently, it is not very clear if the statements say that autoregressive training is sufficient for achieving level-1 causal reasoning.

* Equation 2 removes the dependency over the confounding variable using the total probability formula, where the $P_C$ is the general world knowledge base that may not always be available. Moreover, even if the general knowledge base is available, applying the total probability formula would need it to be complete, covering all the possibilities, which may not be feasible in general for natural language descriptions. I may have understood it incompletely, but it would be good if you could share some more thoughts and assumptions behind equation 2 to make it more transparent to the reader.

* The use of GPT 3.5 turbo for constructing the benchmark may add a bias in the dataset construction. It would be good to highlight such biases and talk about them in detail.


Minor Suggestions:
* The use of fancy terms like AGI would be better avoided since the definitions of these terms are unclear and still under construction.

**Limitations:**

The limitations of the work are highlighted in the appendix.

---

> ### Author Rebuttal · Authors · 2024-08-07
>
> Thank you for  your constructive comments and they are valuable for improving our paper. In the following, I will address your concerns one-by-one.
>
> > Q1. Weakness (1)
>
> A1. Thank you for point out this ambiguity. Here we provide more clear definitions for two causal reasoning levels.
> - **Level-1** causal reasoning refers to retrieving existing causal knowledge embedded in model parameters and the given contexts. It is usually fast and suitable for handling simple causal relationships. Current LLMs are at this level.
> - **Level-2** causal reasoning mimics human cognition, allowing LLMs to use powerful reasoning mechanisms and existing knowledge to infer causal relationships, even with unfamiliar tasks. This process is slower but capable of handling unknown causal knowledge.
>
> We think that level-2 reasoning is not always better, as it is less efficient and cost-effective. Ideally, LLMs would adaptively choose the appropriate reasoning mode based on task difficulty. However, our paper shows that LLMs lack level-2 causal reasoning capabilities, preventing this adaptability.
>
> We have added a link in lines 48-51 to the formal definitions of level-1 and -2 causal reasoning, i.e., Def. 2 and Def. 3.
>
> > Q2. Weakness (2)
>
> A2. We get your concern and clarify it for you. Our paper focuses on **textual causal reasoning tasks**, instead of the numerical ones like classical causal inference/discovery. There is a significant gap in addressing textual tasks with numerical causal methods. Instead, **G$^2$-Reasoner is specifically designed for textual causal reasoning of LLMs** with clear motivations:
> - In Sec. 4, we showed that sequential causality differs from logical causality in natural language, a similar view also proposed by philosopher David Hume [1]. As the autoregressive nature of LLMs, they inherently learn sequential causality. To address this, we proposed a **goal-driven prompt** to steer LLMs in identifying correct cause-effect pairs and reaching the intended goal during decoding.
> - In Sec. 5, we model the data generation process of a textual causal reasoning task with a SCM (Figure 4). This SCM shows that complete **general knowledge essentially drives  causal relationships**. Without it, LLMs may contradict objective causal laws, rendering final reasoning conclusions meaningless.
>
> Thus, G$^2$-Reasoner's performance relies on an general knowledge base. The reported results were obtained using a **very small knowledge base (around 16 MB), yet G$^2$-Reasoner generally achieved non-marginal improvements**. If we use a complete one, such as Wikipedia API, performance can be boosted a lot. However, due to resource constraints, we couldn't repeat all experiments with Wikipedia API. Instead, we are creating a more complete offline general knowledge base and will open source it, which is helpful for LLM causal reasoning and other fields.
>
> In addition to LLM's causal reasoning, we also believe that these two motivations are helpful for general LLM reasoning tasks.
>
> [1] David Hume. A treatise of human nature. Clarendon Press, 1896.
>
> > Q3. Weakness (3)
>
> A3. The 'context' in CausalProbe 2024 **is not a technical contribution for boosting performance but rather a contribution in terms of more reasonable evaluation benchmarks**. Previous causal Q&A benchmarks did not provide contexts. However, we have figured out that LLM's causal reasoning may be meaningless without specific contexts. Figure 5 & 6 exactly highlight this contribution. The causal insights behind G$^2$-Reasoner is discussed in Q2. We also think that G$^2$-Reasoner and necessary contexts will be applicable for non-causal tasks of LLMs.
>
> > Q4. Question (1)
>
> A4. Sorry, we don't fully understand your concerns. For widely-used decoder-only LLMs, the masked self-attention naturally determines that the current token is only influenced by the previous ones and the prompt. We are happy to further discuss this interesting problem with you.
>
> > Q5. Question (2)
>
> A5. Thank you for your insightful comment. We understand your desire for better metrics to measure text complexity. However, our benchmarks are in the form of multiple-choice questions, and the **final outputs are only option IDs**, making it difficult to introduce other metrics.
>
> >Q6. Question (3)
>
> A6. To reach this main conclusion, we constructed CausalProbe 2024, which **has a format and difficulty similar to previous benchmarks** but has a **completely new corpus (see Tables 1 and 3)**, which was released later than the LLMs' data cut-off time. Instead, the previous benchmarks may have been training data. It allows CausalProbe 2024 to test an LLM's **true** causal reasoning ability, and the effect of the LLM's memorized knowledge are partially exluded. Figure 1(d) shows that four LLMs perform significantly worse on CausalProbe 2024 than the previous benchmarks, indicating that autoregressive LLMs only achieve level-1 causal reasoning but not level-2.
>
> > Q7. Question (4)
>
> A7. Your understanding for the Eq. (2) is correct. Ideally, a complete general knowledge base $P_C$ is needed to ensure the total probability formula holds. However, obtaining an absolutely complete $P_C$ is challenging, limiting G$^2$-Reasoner's performance as discussed in Q2.
>
> > Q8. Question (5)
>
> A8. Thank you for your insightful feedback on the potential biases introduced by using GPT-3.5 Turbo to construct our benchmark. We discuss it here:
> - **Model bias**: GPT-3.5 Turbo may inherit biases and limitations from its training data.
> - **Generation bias**: The model may frequently produce certain types of questions or answers.
> - **Language and cultural bias**: Using English news corpora may introduce a Western or Anglophone bias.
>
> We will include this discussion in Sec. 5 of our paper.
>
> > Q9. Minor Suggestions
>
> A9. Thank you for your valuable suggestion. 'AGI' is an unclear concept and using it in a research paper is inappropriate. We used 'AGI' three times and have replaced them with more conservative terms.

---

> ### Author Response · Authors · 2024-08-10
>
> Dear reviewer ZNie,
>
> Thanks for taking the time to review our work. We have carefully considered your comments and made every effort to respond to your concerns.
>
> If you have any further questions or require additional clarification, please kindly let us know.
>
> Best regards,
>
> Authors of Paper 8470

---

> > ### Author Response · Authors · 2024-08-12
> >
> > Dear reviewer,
> >
> > Thank you for your time and effort in reviewing our work. We hope our response has adequately addressed your concerns. If you have any further questions or require additional clarification, please don't hesitate to let us know.
> >
> > Best regards,
> >
> > Authors of Paper 8470

---

> > > ### Comment · Reviewer_ZNie · 2024-08-12
> > >
> > > Dear Authors,
> > >
> > > Thank you for your detailed rebuttal and clarifications. I acknowledge that I have gone through the detailed reviews provided by the reviewers as well as the responses for each of the reviews. Given the length of the discussions, it took some time to go through each of the comments and the responses. I am sorry if there was some delay from my end. The rebuttal helps clarify a few of the concerns. Overall, the formulation for Eq. (2) remains the primary concern and the limitation of the work. As agreed by the authors, it becomes challenging to ensure that the total probability holds. Some of the concreteness of the definition of level-1, and level-2 remains too generic. Please find the comments related to the definition and the clarification regarding one of the questions below:
> > >
> > > The provided definitions for level 1 and level 2 are still not completely concrete and there is a huge spectrum of ambiguity that can exist in the statements. When the authors say “mimics human cognition”, it is difficult to comprehend which specific part of cognition is referred to by the authors. Though they provide a brief intuition about the proposed idea, considering them as definitions may not be correct as clear distinctions cannot be made by the reader.
> > >
> > > Q4. Question (1)) Sorry if the comment was not clear. I want to reiterate that when you say “the current value is not related to future values”, it might not be completely correct as the current values may depend on the future words in any language and similar instances across the dataset will be present, making model learn the relationship between the tokens rather than just depending on the previous tokens. For example, the same sentence can be written in different word orders, “Rome is the capital of Italy” and “The capital of Italy is Rome“, similarly The box contains the red beads” and “The red beads are inside the box”. When trained on large-scale datasets, the occurrence of multiple such instances will make the model learn the relationships between the entities rather than just the tokens in the past. So the assumption that “the current value is **not related** to future values” may not hold completely.
> > >
> > > To summarize, I have raised the initial score after the rebuttal and it would be good if the authors could make the suggested changes by incorporating more discussions as highlighted by the authors in the rebuttal.

---

> > > > ### Author Response · Authors · 2024-08-13
> > > >
> > > > Dear Reviewer ZNie,
> > > >
> > > > Thank you for your reply. We sincerely appreciate your support for our paper and are particularly grateful for your constructive comments. We are happy to see that our response has addressed some your concerns. In the following, we will make every effort to address the remaining concerns.
> > > >
> > > > > Q1. The formulation for Eq. (2) remains the primary concern and the limitation of the work.
> > > >
> > > > A1. We acknowledge your comment about the guarantee of Eq. (2) and agree with its restriction. One of our main contributions is **identifying the importance of general knowledge for LLM causal reasoning, which is usually overlooked before**. However, the realization of our method relies on such a **complete** general knowledge base, but there is **no available one** so far. To mostly fill this gap, we have **two possible options**: **1)** We are making a general knowledge base that is significantly more complete than the one we used. Regarding its heavy work, we will open source it as soon as possible.  **2)** If accessible, another good alternative is the Wikipedia API.
> > > >
> > > > > Q2. The provided definitions for level 1 and level 2 are still not completely concrete and this is a huge spectrum of ambiguity that can exist in the statements.
> > > >
> > > > A2. Thank you for your invaluable feedback. We acknowledge that our use of the phrase "mimics human cognition" was imprecise and not suitable for a rigorous definition. We appreciate your suggestion to refine our description of level-2 causal reasoning regarding **pre-existing and new causal knowledge**. We propose the following revised definition:
> > > >
> > > > *Level-2 causal reasoning refers to the ability of LLMs to **infer new causal knowledge** by leveraging their internal parametric knowledge in combination with provided context, going beyond mere retrieval and simple combination of existing causal knowledge.*
> > > >
> > > > We believe this new perspective offers a more concrete and precise definition than our previous formulation. It emphasizes the LLMs' capacity to generate new causal insights rather than simply retrieving or combining pre-existing knowledge.
> > > >
> > > > > Q3. "the current value is not related to future values" may not be completely correct.
> > > >
> > > > A3. Thank you for this insight. We apologize for the confusion earlier. When we say "the current value is not related to future values", we are referring to the autoregressive mechanism of LLMs. During decoding, the masked self-attention forces the model to rely solely on past tokens for generation. However, your profound insight has reminded me of an important point. Despite this mechanical limitation, due to the diversity of massive training data, LLMs may take in various expressions of the same semantic content, thereby implicitly learning causal relationships between entities, although this process may not be interpretable. We will **weaken our previous statement** in the paper, limiting it to the mechanism of LLMs, and **thoroughly discuss the insight you've brought up**.
> > > >
> > > > **The opportunity to discuss our work with you has been immensely constructive, leading to substantial improvements in our research.** Should you have any further suggestions or comments, please do not hesitate to share them with us. Thank you once again for your invaluable contribution to the enhancement of our research.
> > > >
> > > > Best regards,
> > > >
> > > > Authors of Paper 8470

---

### Author Rebuttal · Authors · 2024-08-07

Here, we mainly present the work we **have done** and the **new efforts** added for data quality control.

> Current quality control for CausalProbe 2024.

We discuss our efforts for ensuring the benchmark's quality here, which have not been discussed in detail in our paper. We have merged them into our paper. Our efforts are threefold:
- **Preparation**: Our corpus is sourced from two famous media with high quality. We further performed an initial cleaning of the corpus using regular expressions. Subsequently, we used the Google DLP API to detect sensitive information (such as pornography, violence, advertisements, etc.) in the corpus and removed any violating content.
- **Production**: We use GPT 3.5 Turbo, one of the best LLMs at the time, to construct the benchmark from the prepared corpus.  To improve its quality, we experimented with different prompt templates and adopted the best.
- **Verification**: First we used Python scripts to exclude incomplete/garbled items. Then we re-organize them to .json format for ease of reading. Finally we went through all items to find out problematic ones and excluded them.

> Newly added quality control.

We have performed **additional quality control** following reviewer ZgYU's comment. This quality control followed a crowdsourcing pipeline. We recruited 17 volunteers, all of whom hold a master’s degree or higher and are currently engaged in frontline research. Additionally, all volunteers are fluent in English. In the preliminary tests, we randomly sampled 20 questions from CausalProbe-H and asked each volunteer to answer them. We then recorded each volunteer's answer and their perceived difficulty level about this task (on a scale of 1-10, with 10 being the most difficult). The selection criteria required the perceived difficulty level to be **no more than 7** and the accuracy rate to be **no less than 80\%**. Ultimately, **13 out of the 17 volunteers** met these criteria and were selected as qualified, and the test results are shown in Table 1 in PDF.

Given the limited time available during the author response phase, we performed quality control on a subset of CausalProbe 2024. Specifically, we **randomly sampled 260 questions** from CausalProbe-H and assigned each question to 3 volunteers randomly, using the Algorithm 1 in PDF. Each volunteer received a total of 60 questions. After receiving their feedback, we treated those questions correctly answered by no less than two volunteers as high-quality data.  Finally, 232 out of 260 questions were filtered out (temporarily called CausalProbe-HQ), achieving **the qualification rate of 89.2\%**. The randomly sampled data IDs and the high-quality ones among them is shown in Table 5 in PDF. Next, we use CausalProbe-HQ to evaluate four LLMs used in our paper again, whose results are shown in Table 2 in PDF. The results show that all four LLMs still perform poorly on this subset, suggesting that their failure is primarily due to **limited causal reasoning abilities rather than the errors in the benchmark**.

We will continue to perform quality control on the remaining data and eventually open-source the fully quality-controlled version.

> Newly added indefinite multi-choice version of CausalProbe 2024.

In addition, we also constructed an **indefinite multi-choice version of CausalProbe 2024**, named as **CausalProbe-M**, following reviewer ZgYU's comment. Its corpus is the same as CausalProbe 2024 and we designed a prompt template to generate it using GPT-4o mini. **We have uploaded CausalProbe-M to the anonymous link attached in our paper**. CausalProbe-M consists of 3441 Q&A data. The number of correct options is **indefinite**, ranging from 1 to 4, and the statistic is shown in Figure 1 in PDF. The ratio of query types (cause or effect) is shown in Figure 2 in PDF.

 We also sampled a subset of CausalProbe-M to perform quality control like the above. The **evaluation results** on CausalProbe-M are shown in Table 3 in PDF. All four LLMs experienced a **more significant performance drop** than CausalProbe-E and -H, under the exact matching (i.e., all the correct answers are exactly picked). However, for the partial matching (i.e., missed options are allowed but incorrect options are not allowed), GPT and Claude performed relatively well, achieving near 75% and 85% accuracy rates, respectively. This suggests that **LLMs cannot fully figure out the causal information in each option**, implying their limited causal reasoning abilities from a new perspective. However, it is gratifying that **LLMs make fewer false positive errors**.

---

### Author Response · Authors · 2024-08-14
**A general response by authors**

**We sincerely thank all four reviewers for their thoughtful suggestions on our submission.**

**We received four reviews with  ratings 6,6,6,5. We are glad that all the reviewers have good impressions of our work**, including

- an interesting and timely problem (ZNie, yorJ)
- create a novel benchmark (ZNie, yorJ, ZgYU)
- provide insightful analyses (NPEc, ZNie, yorJ)
- methodological contributions (yorJ, ZgYU)
- comprehensive evaluation (ZNie, ZgYU)
- well-written and good presentation (NPEc, ZNie, yorJ, ZgYU)

**In the author response period, we have provided detailed responses to all the comments and questions point-by-point.** Specifically, we list the major ones:

- enhance the quality of CausalProbe 2024 by crowdsourcing and constructing an indefinite multiple-choice dataset (W2 for ZgYU)
- provide more concrete and precise definitions for level-1 and -2 causal reasoning (W1 for NPEc, W1 for ZNie)
- repeat the experiments of closed-source LLMs to get standard deviation (W4 for ZgYU, W4 for yorJ)
- clarify the technical insights of our method regarding causality (W1 for ZgYU, W3 for ZNie)
- discuss the reported performance and the implementation of RAG (W2 for ZNie, W1 for ZgYU, W2 for yorJ)
- clarify different construction methods of CausalProbe-E and -H (W3 for ZgYU)
- clarify the role of contexts in CausalProbe 2024 (Q2 for ZgYU)
- further explain the meaning of the "exogenous variable" (W2 for NPEc)
- clarify the implication of Eq. (1) and Eq. (2) (W3 for NPEc, Q4 for ZNie)
- discuss the correctness of "the current value is not related to future values" (Q1 for ZNie)
- discuss the potential bias from GPT 3.5 during constructing CausalProbe 2024 (Q5 for ZNie)
- provide a detailed revision plan for refining the expression (Q3, Q6 for ZNie)

which have been merged in our paper.

Besides, we provide
- the newly constructed CausalProbe-M dataset (an indefinite multiple-choice version) in the anonymous repository
- evaluation results of four LLMs on CausalProbe-M (Table 3)
- complete process details of the additional quality control (Table 1,  Table 2, Table 5, Algorithm 1)
- standard deviations of the closed-source LLMs' results (Table 4)
- statistic of CausalProbe-M (Figure 1, Figure 2).

**Once again, we thank the reviewers for their thorough and thoughtful reviews. We are confident that their suggestions have led to a better version of our paper, and we look forward to its potential contribution to the field.**

Best regards,

Authors of Paper 8470

---

### Decision · Program_Chairs · 2024-09-25

**Decision:**

Accept (poster)

**Comment:**

This work explores the question of whether LLM can perform causal reasoning on a human-like level by incorporating contextual information. The work introduces a new benchmark, CausalProbe-2024, which reveals that LLMs struggle with causal reasoning in fresh and unseen contexts.

This work generated a lot of discussions both during the rebuttal phase as well as behind the scenes and all the reviewers agreed on the utility of the work. The reviewers agree that "the new perspectives on how cause-effect relations manifest in real-world data pose an important, but inherently tricky topic to capture". They also agreed on certain areas of improvement such as making the provided definitions for level 1 and level 2 more concrete and not using terms such as AGI which have an unclear defintion (which the authors mention that it has been removed).

I read the paper and overall I think this is a valuable contribution to the growing area in the intersection of causality and LLM. I recommend acceptance.